# *insomniac* links the development and function of a sleep-regulatory circuit

Qiuling Li[1], Hyunsoo Jang[1†], Kayla Y Lim[1‡], Alexie Lessing[1], Nicholas Stavropoulos[1,2]*

[1]Neuroscience Institute, Department of Neuroscience and Physiology, New York University School of Medicine, New York, United States; [2]Waksman Institute, Rutgers University, Piscataway, United States

**Abstract** Although many genes are known to influence sleep, when and how they impact sleep-regulatory circuits remain ill-defined. Here, we show that *insomniac* (*inc*), a conserved adaptor for the autism-associated Cul3 ubiquitin ligase, acts in a restricted period of neuronal development to impact sleep in adult *Drosophila*. The loss of *inc* causes structural and functional alterations within the mushroom body (MB), a center for sensory integration, associative learning, and sleep regulation. In *inc* mutants, MB neurons are produced in excess, develop anatomical defects that impede circuit assembly, and are unable to promote sleep when activated in adulthood. Our findings link neurogenesis and postmitotic development of sleep-regulatory neurons to their adult function and suggest that developmental perturbations of circuits that couple sensory inputs and sleep may underlie sleep dysfunction in neurodevelopmental disorders.

**\*For correspondence:** stavropoulos@waksman.rutgers. edu

**Present address:** [†]Department of Biological Sciences, Korea Advanced Institute of Science and Technology (KAIST), Daejeon, South Korea; [‡]Graduate Program in Molecular, Cellular, & Integrative Physiology, University of California, Los Angeles, United States

**Competing interest:** The authors declare that no competing interests exist.

## Editor's evaluation

This is an interesting study showing that the short sleep phenotype of *inc* mutants in *Drosophila* depends on the loss of the gene at a specific developmental time, and in a specific region, the mushroom bodies (MB). There are very few studies assessing the effects of sleep during development, in any animal species, and thus this paper is a very welcomed addition. The experiments are carefully done, and the conclusions are warranted.

## Introduction

A central goal of sleep research has been elucidating the mechanisms by which genes shape normal sleep patterns and cause sleep disorders. While numerous genes that strongly impact sleep have been identified in humans and in animals ranging from mammals to invertebrates (*Chemelli et al., 1999*; *Chiu et al., 2016*; *Cirelli et al., 2005*; *Funato et al., 2016*; *He et al., 2009*; *Lin et al., 1999*; *Raizen et al., 2008*), when these genes act to influence sleep is in many cases unresolved. Genes that act in the adult brain to modulate the activity of sleep-regulatory circuits in an ongoing manner have been intensively investigated (e.g. *Chemelli et al., 1999*; *Lin et al., 1999*), including with conditional gain-of-function, loss-of-function, and rescue in adult animals (*Chiu et al., 2016*; *Clasadonte et al., 2017*; *Foltenyi et al., 2007*; *Guo et al., 2011*; *Ishimoto and Kitamoto, 2010*; *Joiner et al., 2006*; *Van Buskirk and Sternberg, 2007*). In contrast, despite great progress in understanding neuronal development (*Doe, 2008*; *Jessell and Sanes, 2000*; *Sanes and Zipursky, 2020*; *Tessier-Lavigne and Goodman, 1996*; *Weinstein and Hemmati-Brivanlou, 1999*), developmental mechanisms by which genes influence sleep remain poorly explored, despite the likely relevance of such mechanisms to sleep disturbances in autism and other neurodevelopmental disorders (*Angriman et al., 2015*; *Souders et al., 2017*). Notably, the temporal contributions of genes that impact sleep are rarely

assessed in a comprehensive manner, and a further challenge has been linking particular genes to developmental processes that control the structure and function of discrete sleep-regulatory circuits.

Here, we assess the temporal contributions of *insomniac* (*inc*), a gene whose mutation sharply curtails sleep in *Drosophila* (*Pfeiffenberger and Allada, 2012*; *Stavropoulos and Young, 2011*). Pan-neuronal depletion of *inc* causes short sleep, while restoring *inc* solely to neurons is largely sufficient to rescue the sleep deficits of *inc* mutants, indicating that *inc* impacts sleep chiefly through neurons (*Pfeiffenberger and Allada, 2012*; *Stavropoulos and Young, 2011*). *inc* is expressed in the larval, pupal, and adult brain (*Pfeiffenberger and Allada, 2012*; *Stavropoulos and Young, 2011*), but when *inc* acts to influence sleep remains uncertain (*Li and Stavropoulos, 2016*; *Pfeiffenberger and Allada, 2012*). *inc* encodes an adaptor for the Cul3 ubiquitin ligase (*Li et al., 2019*), which, like *inc*, is required in neurons for normal sleep (*Pfeiffenberger and Allada, 2012*; *Stavropoulos and Young, 2011*). Both *inc* and *Cul3* are highly conserved, and mammalian *inc* orthologs restore sleep to *inc* mutants (*Li et al., 2017*), suggesting that functions and substrates of *inc* are conserved in mammals. Human *Cul3* mutations are implicated as a cause of autism and its associated sleep dysfunction (*Codina-Solà et al., 2015*; *Kong et al., 2012*; *O'Roak et al., 2012*), but the underlying mechanisms are unknown. Studies of *inc* may thus reveal fundamental and conserved mechanisms underlying sleep regulation which are altered in sleep disorders.

Using conditional genetic manipulations of *inc*, we show that *inc* acts transiently in developing neurons to impact sleep in adulthood. We furthermore identify developmental defects in *inc* mutants within the mushroom body (MB), a brain structure that integrates sensory stimuli and regulates sleep. Loss of *inc* alters MB neurogenesis, causing the overproduction of late-born neurons and changes

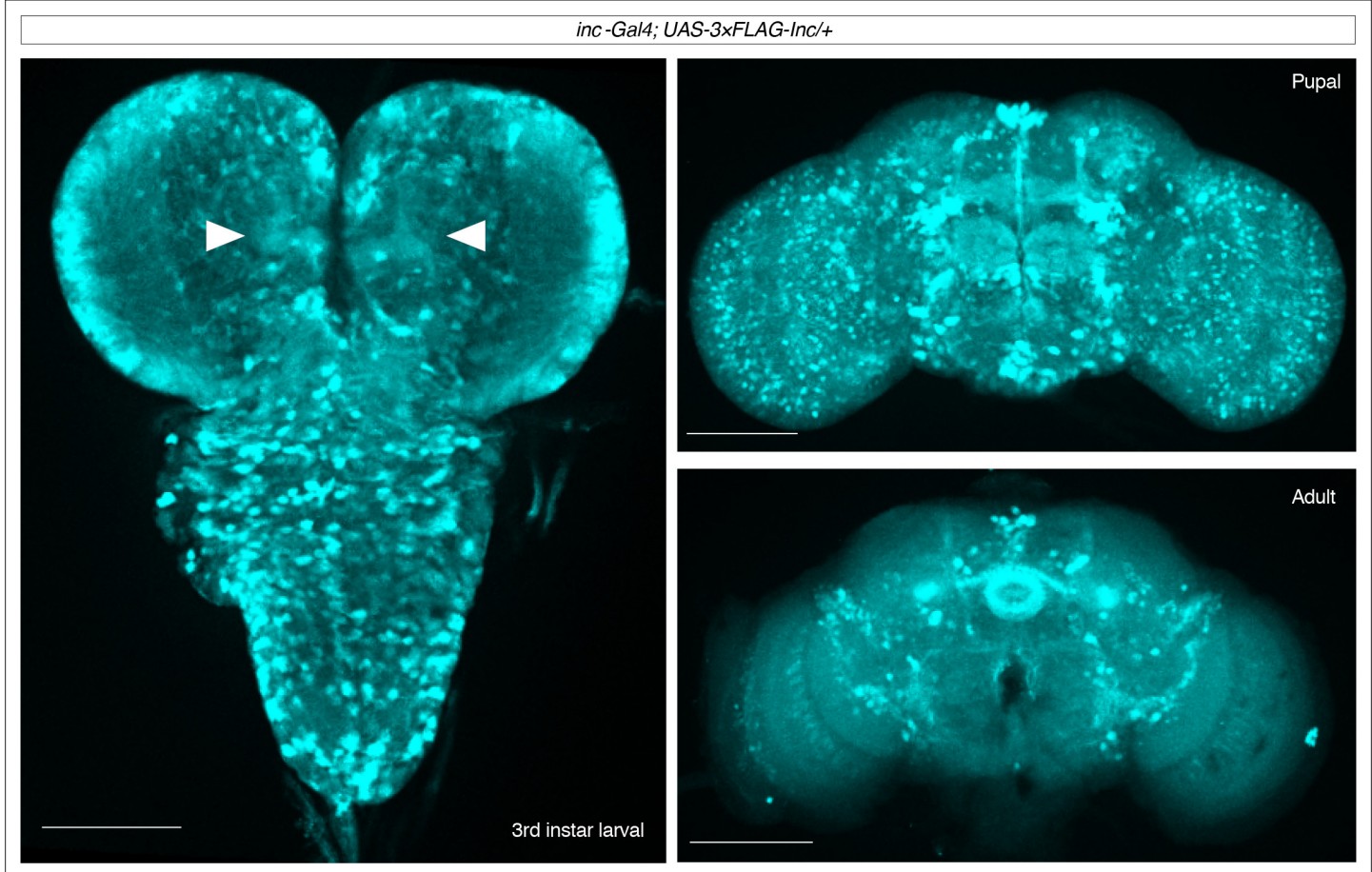

**Figure 1.** Expression of 3×FLAG-*inc* driven by *inc-Gal4* in the larval, pupal, and adult brain. Maximal projections are shown for male *inc-Gal4; UAS-3×FLAG-inc/+* brains stained with anti-FLAG. For larval brain, projection from a partial *z*-stack is shown to allow visualization of signal in mushroom body projections (arrowheads). In pupae and adults, signal is prominent in the mushroom body, pars intercerebralis, fan-shaped body, and ellipsoid body. Scale bars, 100 μm.

in postmitotic development that impair the assembly of MB circuits. These developmental alterations persist into adulthood and are associated with specific deficits in the ability of MB neurons to promote sleep in *inc* adults, in contrast to the anatomy and function of other sleep-regulatory circuits which remain intact. Together, these results elucidate an unexpected mechanism by which *inc* shapes the development and function of sleep-regulatory neurons to exert a lasting impact on sleep–wake behavior. Our findings additionally suggest that developmental alterations of neurogenesis and within brain centers that integrate sensory inputs may contribute to sleep dysfunction in autism and other neurodevelopmental disorders.

## Results

### *inc* acts transiently during a restricted developmental period to impact sleep in adulthood

*inc* impacts sleep through neurons and is expressed in the developing and adult brain (*Figure 1*; *Pfeiffenberger and Allada, 2012*; *Stavropoulos and Young, 2011*). To assess the temporal mechanisms by which *inc* impacts sleep, we manipulated *inc* expression in neurons using the ligand-inducible Q-system (*Potter et al., 2010*; *Riabinina et al., 2015*). The Q-system circumvents nonspecific perturbations of sleep caused by other inducible systems and allows constitutive, developmental, and adult manipulations of sleep (*Li and Stavropoulos, 2016*). We performed a series of conditional rescue experiments in short-sleeping *inc*[1] null mutants bearing a *UAS-inc-HA* transgene whose expression is induced in neurons by the Q-system upon exposure to quinic acid (*Figure 2A*). Animals exposed to vehicle throughout development and adulthood slept indistinguishably from *inc*[1] mutants, while animals exposed constitutively to quinic acid exhibited strongly rescued sleep (*Figure 2B, C*; *Figure 2—figure supplement 1*), consistent with the rescue conferred by constitutive neuronal expression of *inc* (*Pfeiffenberger and Allada, 2012*; *Stavropoulos and Young, 2011*). Anti-HA staining of brains confirmed that the Q-system controlled *inc* expression as expected: vehicle-fed animals lacked *inc*-HA signal, while those exposed constitutively to quinic acid expressed *inc*-HA in the larval, pupal, and adult brain (*Figure 2D*). We next asked whether *inc* influences sleep through adult-specific or developmental mechanisms. Animals fed quinic acid in adulthood expressed *inc*-HA in the adult brain but exhibited no rescue of their sleep deficits (*Figure 2B–D*; *Figure 2—figure supplement 1*). In stark contrast, developmental induction of *inc*-HA from embryonic through pupal stages restored sleep to near wild-type levels (*Figure 2B–D*; *Figure 2—figure supplement 1*). These findings indicate that *inc* is dispensable in adult neurons and acts instead during neuronal development to ultimately impact sleep–wake behavior.

We further defined the developmental period in which *inc* functions, using more precise temporal manipulations. Neuronal induction of *inc*-HA from the late third instar larval stage through adulthood strongly rescued the *inc* sleep phenotype (*Figure 2B–D*; *Figure 2—figure supplement 1*), indicating that *inc* is dispensable in embryonic and early larval neurons. Induction of *inc* activity solely in late third instar larval and pupal neurons, using a pulse of quinic acid exposure (*Figure 2D*), restored sleep indistinguishably from constitutive neuronal induction (*Figure 2B, C*; *Figure 2—figure supplement 1*). The sleep deficits of *inc*[2] animals, which bear an independent *inc* null allele that can be reverted by Gal4 (*Stavropoulos and Young, 2011*), were similarly rescued by this pulse of quinic acid (*Figure 3A–C*; *Figure 3—figure supplement 1*), confirming that *inc* activity in this developmental period is sufficient to restore sleep to *inc* mutants. We next assessed whether *inc* is required in late third instar larval and pupal neurons for normal sleep in adulthood, by using the Q-system to induce a pulse of *inc* RNAi. This manipulation markedly decreased sleep (*Figure 3D, E*; *Figure 3—figure supplement 2*). Together, these findings indicate that *inc* acts transiently in neurons of late third instar larvae and pupae to influence adult sleep–wake behavior. During these developmental stages, many neurons of the adult brain are born and assemble into circuits (*Truman and Bate, 1988*; *White and Kankel, 1978*).

### *inc* has a critical function in the MB that impacts sleep

To identify neurons that might underlie the developmental impact of *inc* on sleep, we performed a rescue screen in *inc*[2] mutants. We screened 277 Gal4 lines expressed in sleep-regulatory circuits or randomly selected populations of cells in the brain and identified two drivers, *c253-Gal4* and *c309-Gal4*, that rescued sleep similarly to the pan-neuronal *nsyb-Gal4* driver (*Figure 4A*). After backcrossing

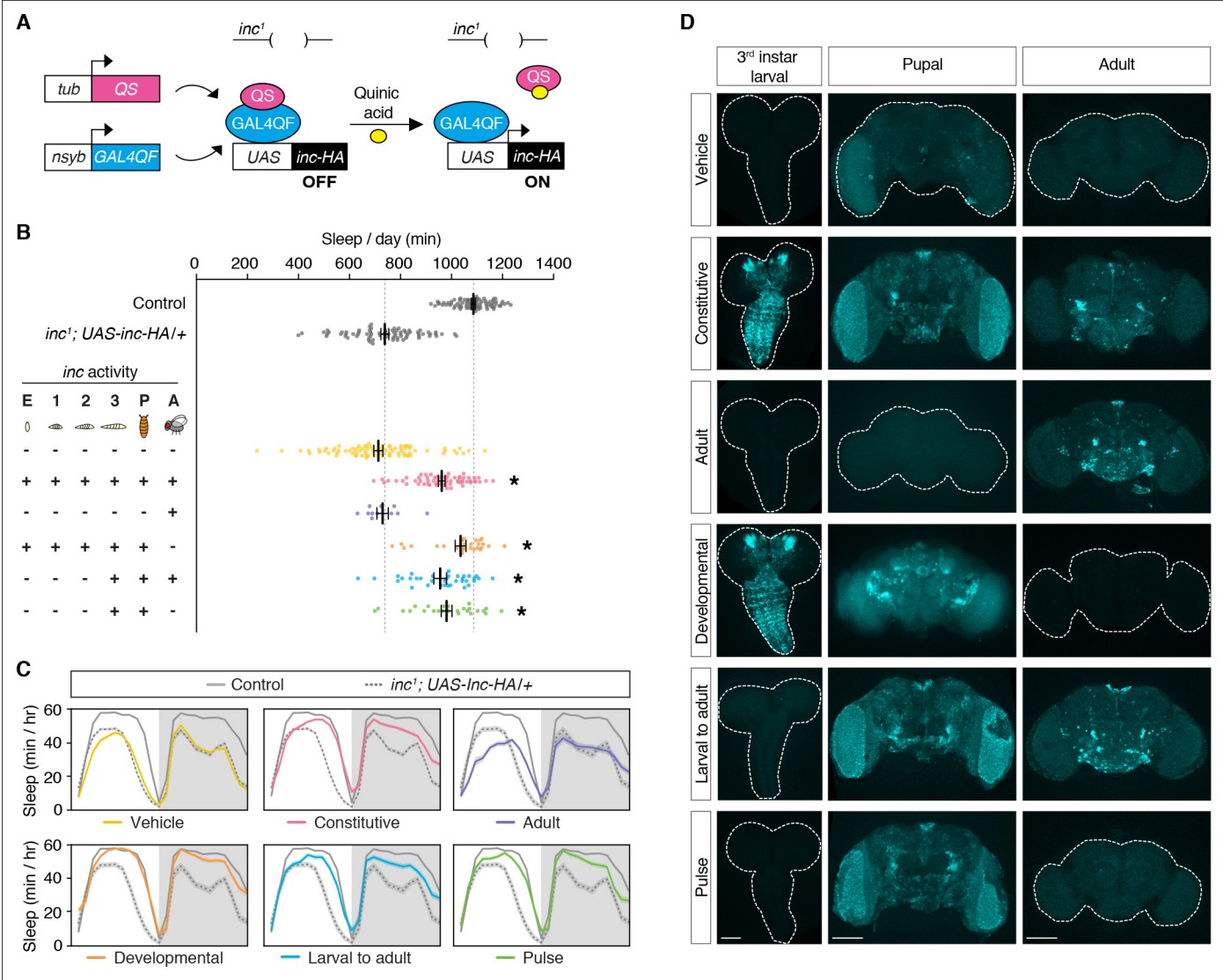

**Figure 2.** *inc* acts in a restricted period of neuronal development to impact sleep in adulthood. (**A**) Conditional rescue of *inc¹* mutants using the ligand-inducible Q-system. Quinic acid relieves QS suppression of the pan-neuronally expressed Gal4QF transcriptional activator, inducing *UAS-inc-HA* in neurons. (**B**) Total sleep duration of controls (gray) and *inc¹; UAS-inc-HA/tub-QS; nsyb-GAL4QF/+* animals exposed to quinic acid (+) or vehicle (−) at indicated life stages; embryos (E), larval stages (1–3), pupae (P), and adults (A). Bars represent mean ± standard error of the mean (SEM). *n* = 11–86. One-way analysis of variance (ANOVA) (*F*(7,397) = 86.73, p < 0.0001) and Tukey post hoc tests, *p < 0.01 for comparisons to *inc¹; UAS-inc-HA/+*. (**C**) Average sleep profiles of flies in (**B**), with induction regimens indicated below. Shading indicates ± SEM. (**D**) Anti-HA staining of *inc¹; UAS-inc-HA/tub-QS; nsyb-GAL4QF/+* brains from indicated induction regimens. Scale bars, 100 μm.

The online version of this article includes the following figure supplement(s) for figure 2:

**Figure supplement 1.** Additional sleep parameters for conditional rescue of *inc¹* mutants using the Q-system.

to an isogenic background, both drivers retained their ability to rescue most of the sleep phenotypes of *inc²* mutants (**Figure 4B, C**; **Figure 4—figure supplement 1**). In late third instar larvae and adults, *c253-Gal4* and *c309-Gal4* are strongly expressed in the MB (**Figure 4D**), a structure important for sensory integration, associative learning, and sleep regulation (**Heisenberg, 2003**; **Joiner et al., 2006**; **Pitman et al., 2006**). Because *c253-Gal4* and *c309-Gal4* are also expressed outside of the MB, we used independent genetic manipulations to confirm that *inc* acts in the MB to influence sleep. *inc-Gal4*, a driver that bears *inc* regulatory sequences and fully rescues *inc* mutants when used to restore *inc* activity (**Li et al., 2017**; **Stavropoulos and Young, 2011**), is expressed in the larval, pupal, and adult MB (**Figure 1**). We tested whether the rescue conferred by *inc-Gal4* was altered by *MB-Gal80*, a Gal4 suppressor expressed in MB neurons during development and adulthood (**Krashes et al., 2007**;

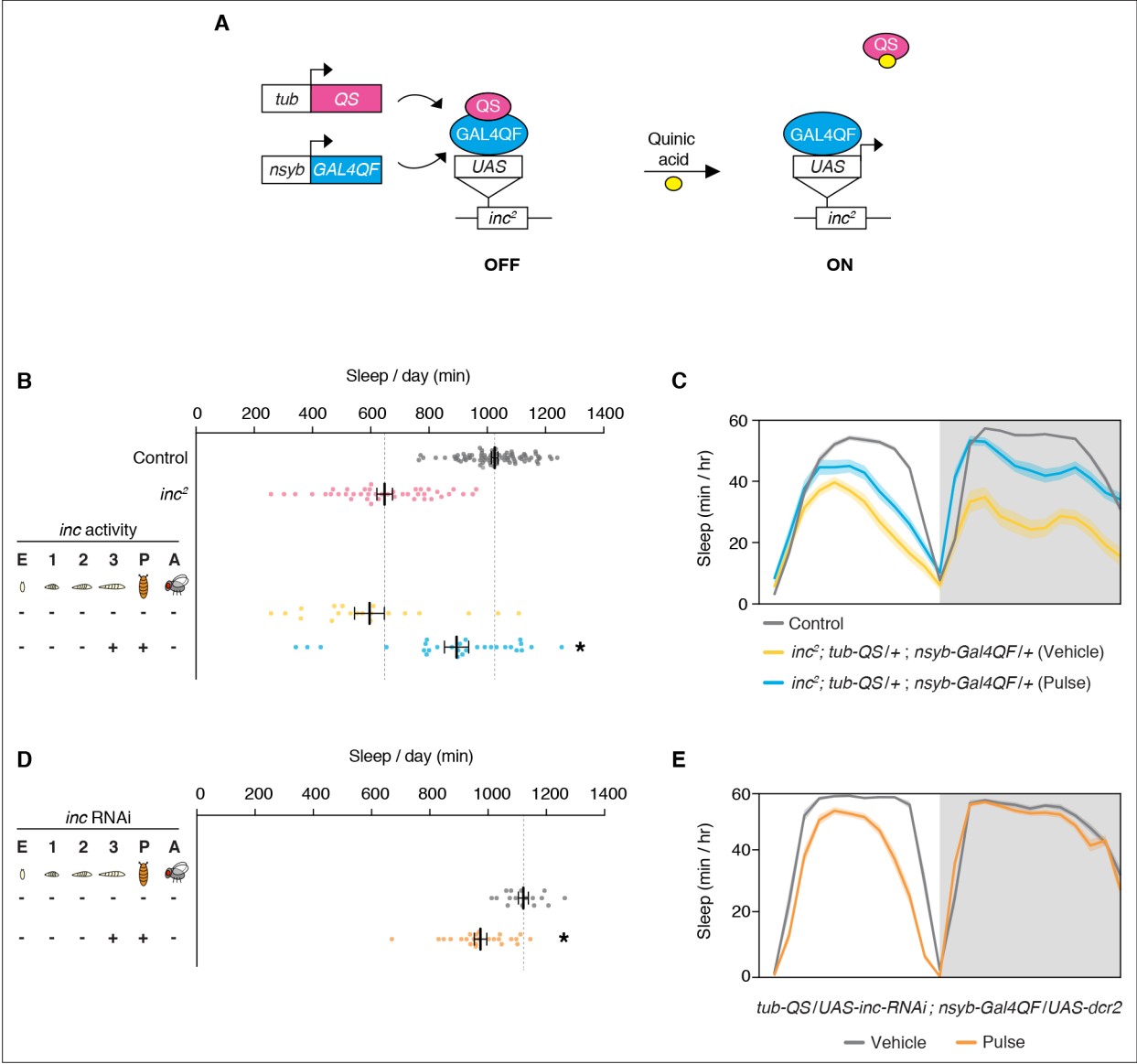

**Figure 3.** Conditional rescue of *inc²* mutants and conditional *inc* RNAi in larval and pupal neurons. (**A**) Conditional neuronal rescue of *inc²* mutants using the ligand-inducible Q-system. *inc²* mutants contain a transposon insertion in the *inc* 5′UTR immediately upstream of the endogenous start codon. A UAS/TATA element within the transposon terminus permits Gal4-dependent restoration of *inc* expression (**Stavropoulos and Young, 2011**). (**B**) Total sleep duration in *inc²; tub-QS/+; nysb-Gal4QF/+* animals exposed to vehicle or quinic acid at the late third instar larval and pupal stages. *n* = 20–83. One-way analysis of variance (ANOVA) ($F_{(3, 170)}$ = 70.66, p > 0.0001) and Tukey post hoc tests, *p < 0.01 for comparisons to *inc²*. (**C**) Average sleep profiles of indicated genotypes from (**B**). (**D**) Total sleep duration in *tub-QS/UAS-inc-RNAi; nsyb-Gal4QF/UAS-dcr2* animals exposed to vehicle or quinic acid at the late third instar larval and pupal stages. *n* = 16–24. Student's *t*-test, *p < 0.01 for comparison to vehicle-treated control. (**E**) Average sleep profiles of animals from (**D**). For (**B**) and (**D**), bars represent mean ± SEM. For (**C**) and (**E**), shading represents ± SEM.

The online version of this article includes the following figure supplement(s) for figure 3:

**Figure supplement 1.** Additional sleep parameters for conditional *inc²* rescue in third instar larval and pupal neurons.

**Figure supplement 2.** Additional sleep parameters for conditional *inc* RNAi in larval and pupal neurons.

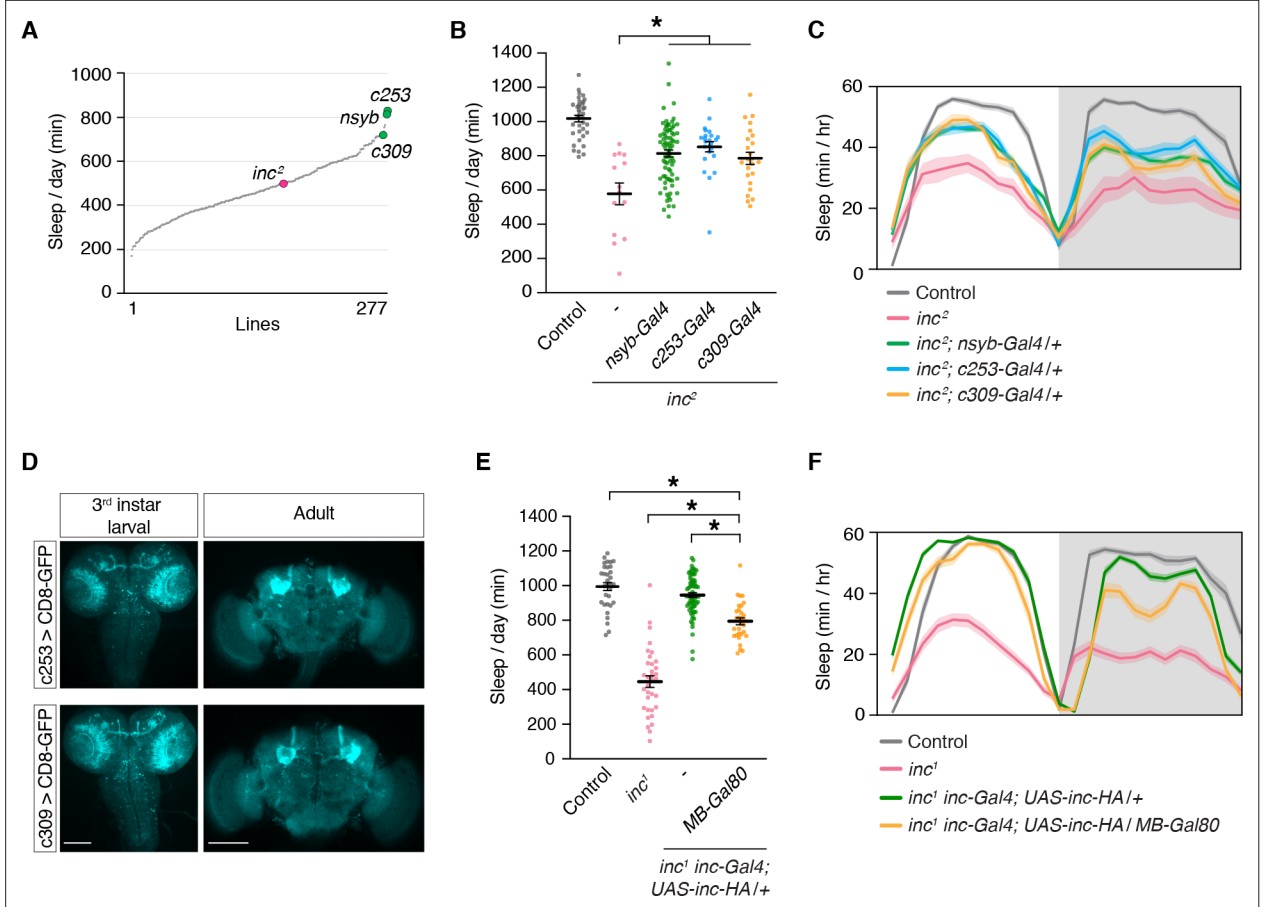

**Figure 4.** The mushroom body is a critical brain region through which *inc* impacts sleep. (**A**) Mean sleep is plotted for each line in a Gal4 rescue screen of *inc²* animals. n ≥ 5 per genotype. (**B**) *c253-Gal4* and *c309-Gal4* rescue sleep in *inc²* mutants. n = 14–78. One-way analysis of variance (ANOVA) ($F(4, 172) = 20.36$, $p < 0.0001$) and Tukey post hoc test, *$p < 0.01$ for comparisons to *inc²*. (**C**) Average sleep profiles of flies in (**B**). (**D**) Anti-GFP immunostaining of indicated genotypes. Scale bars, 100 μm. (**E**) *MB-Gal80* suppresses sleep rescue in *inc¹ inc-Gal4; UAS-inc-HA/+* animals. n = 30–69. One-way ANOVA ($F(3, 161) = 121.4$, $p < 0.0001$) and Tukey post hoc tests, *$p < 0.01$. (**F**) Average sleep profiles of indicated genotypes from (**E**). For (**B**) and (**E**), bars represent mean ± SEM. For (**C**) and (**F**), shading represents ± SEM.

The online version of this article includes the following figure supplement(s) for figure 4:

**Figure supplement 1.** Rescue of *inc* sleep phenotypes by *c253-Gal4* and *c309-Gal4*.

**Figure supplement 2.** Additional parameters for suppression of sleep rescue in *inc¹ inc-Gal4; UAS-inc-HA* animals by *MB-Gal80*.

*Pauls et al., 2010*). *MB-Gal80* partially suppressed the ability of *inc-Gal4* to restore sleep to *inc¹* mutants, indicating that while *inc* does not influence sleep solely through the MB, *inc* is required in MB neurons for normal sleep regulation (*Figure 4E, F*; *Figure 4—figure supplement 2*).

## Loss of *inc* abolishes the sleep-promoting functions of MB neurons but spares the functions of other sleep-regulatory circuits

While different circuits within the MB can promote or inhibit sleep upon activation (*Joiner et al., 2006*; *Pitman et al., 2006*; *Sitaraman et al., 2015a*), ablation of the MB strongly reduces sleep (*Joiner et al., 2006*; *Pitman et al., 2006*), suggesting that the integrated activity of the MB is sleep-promoting. To assess whether the sleep-regulatory functions of the MB are altered in *inc* mutants, we activated MB neurons in adult wild-type and *inc¹* flies using the dTrpA1 heat-activated cation channel (*Hamada et al., 2008*). Wild-type control flies lacking Gal4 drivers exhibited no change in total sleep when shifted to 28.5°C for 24 hr, while *inc¹* flies lacking Gal4 drivers exhibited decreased sleep at this temperature (*Figure 5A, B*), suggesting that *inc* mutants are hyperarousable by thermal stimuli, as for mechanical stimuli (*Pfeiffenberger and Allada, 2012*). Activation of neurons expressing TrpA1 under

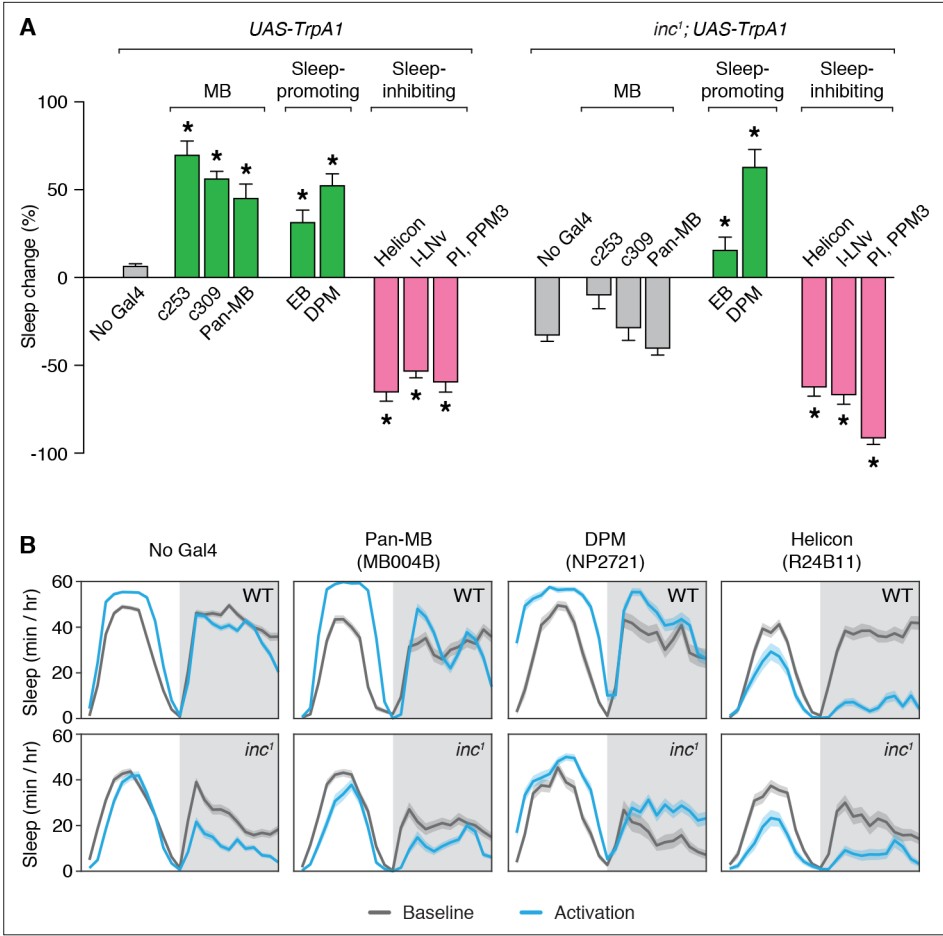

**Figure 5.** Sleep-promoting functions of the mushroom body are impaired in *inc* mutants. (**A**) Thermogenetic activation of neuronal populations expressing TrpA1 in control and *inc¹* animals. Percent change in sleep (mean ± SEM) elicited by activation is shown. n = 31–144. Control and *inc¹* animals expressing dTrpA1 are compared to no-Gal4 controls (*UAS-dTrpA1/+* and *inc¹; UAS-dTrpA1/+*, respectively). *p < 0.01 for Dunnet's post hoc comparisons after one-way analysis of variance (ANOVA) for control animals ($F_{(8, 528)}$ = 92.12, p < 0.0001) or *inc¹* mutants ($F_{(8, 452)}$ = 50.01, p < 0.0001). Green and pink bars indicate drivers that significantly promote or inhibit sleep, respectively; gray bars indicate no significant change with respect to controls. (**B**) Average sleep profiles of animals from (**A**) on the baseline day and during thermogenetic activation. Shading represents ± SEM.

The online version of this article includes the following figure supplement(s) for figure 5:

**Figure supplement 1.** Sleep profiles during baseline and neuronal activation.

**Figure supplement 2.** A split-Gal4 driver expressed specifically in mushroom body (MB) neurons.

the control of *c253-Gal4* or *c309-Gal4* strongly increased sleep in wild-type animals (*Figure 5A, B*; *Figure 5—figure supplement 1*), consistent with observations that inactivating synaptic output using the same drivers promotes wakefulness (*Pitman et al., 2006*). Because *c253-Gal4* and *c309-Gal4* are expressed in some cells outside of the MB, we also assessed a split-Gal4 driver expressed specifically in MB neurons (*Figure 5—figure supplement 2*). Using this driver to express TrpA1 and activate MB neurons increased sleep in wild-type animals (*Figure 5A, B*, 'pan-MB'). Strikingly, using the same three drivers to activate neurons in *inc¹* mutants elicited no significant changes in sleep compared to *inc¹; UAS-TrpA1/+* controls (*Figure 5A, B*; *Figure 5—figure supplement 1*), indicating that the sleep-promoting effects of MB activation are abolished in *inc* mutants.

To test whether the loss of *inc* specifically impairs the sleep-regulatory functions of MB neurons or causes more general deficits in sleep regulation, we assessed other neuronal populations that influence sleep. Activation of sleep-promoting populations that include ellipsoid body R5 (EB) (*Liu et al., 2016*) or Dorsal Paired Medial (DPM) neurons (*Haynes et al., 2015*) increased sleep similarly in

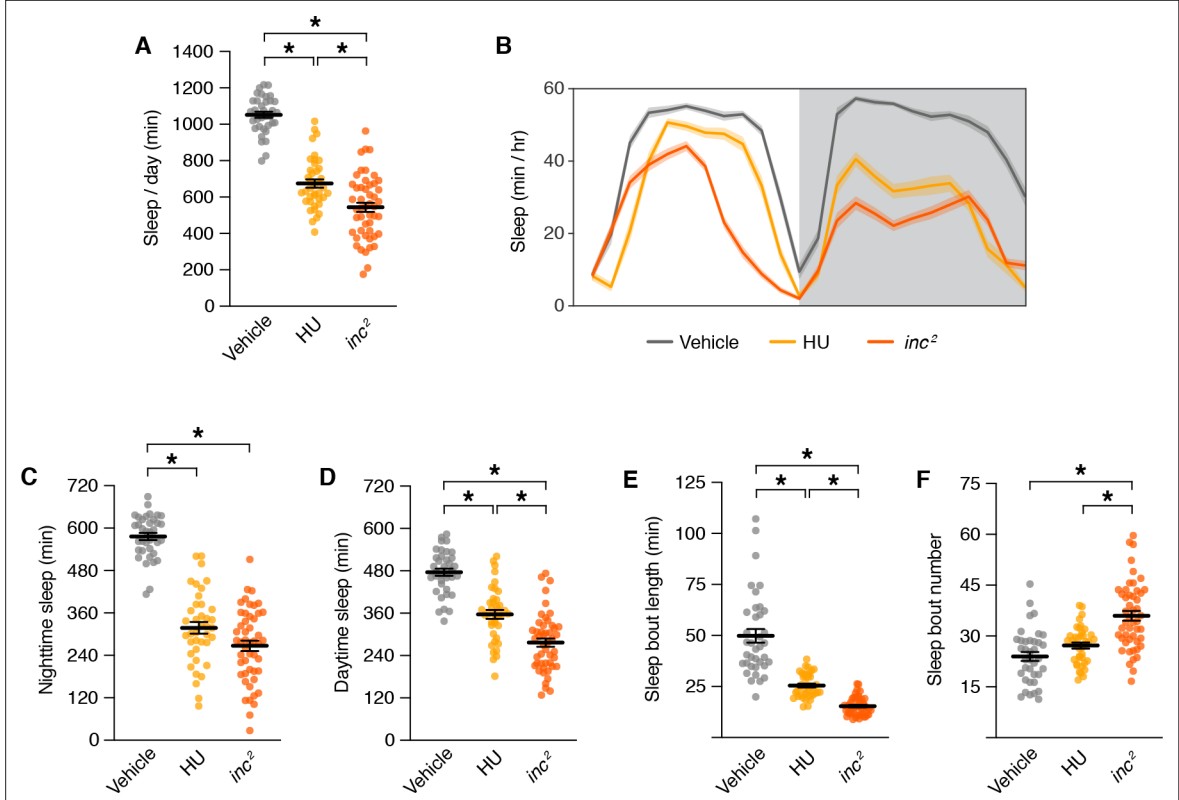

**Figure 6.** Sleep phenotypes for mushroom body ablation and *inc* mutants. Sleep parameters for *inc²* mutants and animals exposed to vehicle or hydroxyurea (HU). For all panels, $n$ = 37–49; *$p < 0.01$ for post hoc tests. (**A**) Total sleep. One-way analysis of variance (ANOVA) ($F_{(2, 122)}$ = 132.9, $p < 0.0001$) and Tukey post hoc tests. (**B**) Average daily sleep profiles. Shading represents ± SEM. (**C**) Nighttime sleep. One-way ANOVA ($F_{(2, 122)}$ = 126.6, $p < 0.0001$) and Tukey post hoc tests. (**D**) Daytime sleep. One-way ANOVA ($F_{(2, 122)}$ = 74.32, $p < 0.0001$) and Tukey post hoc tests. (**E**) Sleep bout length. Kruskal–Wallis ($p < 0.0001$) and Dunn's post hoc tests. (**F**) Sleep bout number. One-way ANOVA ($F_{(2, 122)}$ = 24.89, $p < 0.0001$) and Tukey post hoc tests. For (**A**) and (**C–F**), bars represent mean ± SEM.

wild-type and *inc¹* animals (***Figure 5A, B***; ***Figure 5—figure supplement 1***). Conversely, activation of sleep-inhibiting populations that include Helicon (***Donlea et al., 2018***), l-LN$_v$ (***Sheeba et al., 2008***), or pars intercerebralis and dopaminergic PPM3 neurons (PI, PPM3) (***Dubowy et al., 2016***) strongly decreased sleep in wild-type and *inc¹* animals (***Figure 5A, B***; ***Figure 5—figure supplement 1***). The functions of these populations thus appear to be intact in *inc* mutants, suggesting that the loss of *inc* specifically impairs the sleep-regulatory functions of MB neurons. These findings, together with the developmental time-of-action of *inc* and its requirement within the MB for normal sleep, suggest that *inc* acts developmentally in MB neurons to have a lasting impact on their sleep-regulatory functions in adulthood.

### *inc* regulates the production and anatomy of late-born MB neurons

During the critical developmental period through which *inc* impacts sleep, MB neurons are born and assemble into adult circuits (***Ito and Hotta, 1992***; ***Lee et al., 1999***). In each brain hemisphere, four MB neuroblasts proliferate to yield ~2000 neurons comprising seven sequentially born subtypes (γ$_d$, γ$_m$, α′/β′$_{ap}$, α′/β′$_m$, α/β$_p$, α/β$_s$, and α/β$_c$) that project axons into distinct lobes (γ, α′/β′, and α/β) (***Aso et al., 2014a***; ***Ito et al., 1997***; ***Ito and Hotta, 1992***; ***Kurusu et al., 2002***; ***Lee and Luo, 1999***; ***Tanaka et al., 2008***; ***Truman and Bate, 1988***; ***Zhu et al., 2003***). Chemical ablation of the MB by exposing first instar larvae to hydroxyurea, an inhibitor of DNA replication, causes sleep deficits in adulthood (***Joiner et al., 2006***; ***Pitman et al., 2006***). The sleep deficits caused by MB ablation are similar to but less severe than those of *inc* mutants, including reductions in sleep across the day and decreased sleep consolidation (***Figure 6A–F***). These findings and the partial suppression of *inc* rescue by *MB-Gal80*

(*Figure 4E, F*; *Figure 4—figure supplement 2*) support the notion that reduced sleep in *inc* mutants results from impairments in the MB, alongside effects in additional neuronal populations.

To determine whether *inc* mutants have anatomical changes in the adult MB that might disrupt its sleep-regulatory functions, we examined MB neurons expressing *UAS-Myr-GFP-2A-RedStinger*, a bicistronic reporter that marks projections and nuclei (*Daniels et al., 2014*). Specifically, we used split-Gal4 drivers that label MB neuron subtypes born in embryos ($\gamma_d$), late larval stages ($\alpha'/\beta'$), and in pupae ($\alpha/\beta_c$) (*Aso et al., 2014a*), to assess whether the loss of *inc* might preferentially alter subtypes whose birth and development coincides with the critical period through which *inc* impacts sleep. Consistent with this notion, we observed prominent changes in the number and anatomy of larval- and pupal-born MB neurons in *inc* mutants. While embryonic-born $\gamma_d$ neurons were present in similar numbers in adult brains of controls and *inc[1]* mutants (control, 102 ± 4; *inc[1]*, 94 ± 2) (*Figure 7A, B*), the number of larval-born $\alpha'/\beta'$ neurons was increased 58% in *inc[1]* animals (control, 141 ± 13; *inc[1]*, 223 ± 24), and the number of pupal-born $\alpha/\beta_c$ neurons was doubled (control, 223 ± 11; *inc[1]*, 458 ± 45). The surplus of $\alpha'/\beta'$ and $\alpha/\beta_c$ neurons varied between left and right hemispheres in individual *inc[1]* brains and this variation was greatest for $\alpha/\beta_c$ neurons, the last-born in the MB (*Figure 7A, C*; *Figure 7— figure supplement 1*), indicating that *inc* mutants have a stochastic and cumulative defect in MB neurogenesis. Four clusters of $\alpha/\beta_c$ neurons were present in control animals, reflecting their birth from four MB neuroblasts (*Ito et al., 1997*; *Ito and Hotta, 1992*; *Truman and Bate, 1988*), whereas *inc[1]* mutants exhibited an average of nearly seven clusters (control, 3.7 ± 0.2; *inc[1]*, 6.8 ± 0.6) (*Figure 7A, D*; *Figure 7—figure supplement 1*), suggesting an origin from aberrant or excess neuroblasts. The numbers of other sleep-regulatory neurons, including those of the dorsal fan-shaped body (dFB) and DH44[+] neurons, were unchanged in *inc* mutants (*Figure 7A, I*), indicating that neuronal overproduction in *inc* mutants is specific to the MB or manifests preferentially within this neuronal lineage. These findings indicate that *inc* regulates neurogenesis, a fundamental process regulated by proteins conserved from flies to mammals (*Doe, 2008*; *Knoblich, 2008*), and suggest that alterations in early nervous system development can exert a lasting impact on sleep.

To further assess MB anatomy in *inc* mutants, we examined axons marked with myr-GFP and separately examined dendrites by expressing DenMark (*Nicolaï et al., 2010*). Axons of embryonic-born $\gamma_d$ neurons exhibited no obvious changes in *inc[1]* mutants (*Figure 7A, H*). In contrast, axons of larval- and pupal-born MB neurons exhibited morphological defects whose severity correlated with neuronal overproduction and birth order (*Figure 7A, H*). While $\alpha'/\beta'$ axons were absent from MB lobes in a minority (10%) of *inc[1]* brains, axons of $\alpha/\beta_s$ neurons, the penultimate to be born, were missing from MB lobes in 53% of *inc[1]* brains (1.07 ± 0.33 missing lobes per brain) (*Figure 7H*). Axons of last-born $\alpha/\beta_c$ neurons showed the most severe defects; they failed to project into lobes in 86% of *inc* brains (2.23 ± 0.3 missing lobes per brain), fasciculated from ectopic neuronal clusters, and often aggregated near the peduncle (*Figure 7A, H*; *Figure 7—figure supplement 1*). The dendrites of $\gamma_d$, $\alpha'/\beta'$, and $\alpha/\beta_c$ neurons occupied enlarged territories in *inc* mutants but otherwise appeared normal (*Figure 7F, G*). Expansions in dendritic volume for $\alpha'/\beta'$ and $\alpha/\beta_c$ subtypes paralleled increases in the numbers of these neurons (*Figure 7A, B*), while increases for $\gamma_d$ dendrites occurred independently of neuron number, consistent with functions of *inc* in postmitotic $\gamma_d$ neurons or non-cell autonomous mechanisms. Axons and dendrites of other sleep-regulatory circuits, including those of the dFB, CRZ[+] neurons, and PDF[+] circadian pacemaker neurons, exhibited no obvious changes in *inc* mutants (*Figure 7I*; *Figure 7— figure supplement 2*), suggesting that alterations of neuronal anatomy in *inc* mutants are specific to the MB. These findings indicate that increases in the numbers of late-born MB neurons in *inc* mutants are associated with changes in postmitotic development expected to perturb circuit assembly and function. In particular, the altered axons of multiple MB neuron subtypes are unlikely to form normal circuits with their targets that influence sleep, including dopaminergic neurons, MB output neurons, and recurrent connections to the MB (*Aso et al., 2014b*; *Sitaraman et al., 2015a*; *Sitaraman et al., 2015b*).

## Discussion

Here, we have used temporally restricted genetic manipulations to show that *inc* acts during neuronal development to ultimately impact sleep in adulthood. While many genes are known to act in adults to impact sleep, developmental mechanisms underlying sleep regulation have only recently gained attention (*Chakravarti Dilley et al., 2020*; *Gong et al., 2021*; *Iwasaki et al., 2021*; *Xie et al., 2019*).

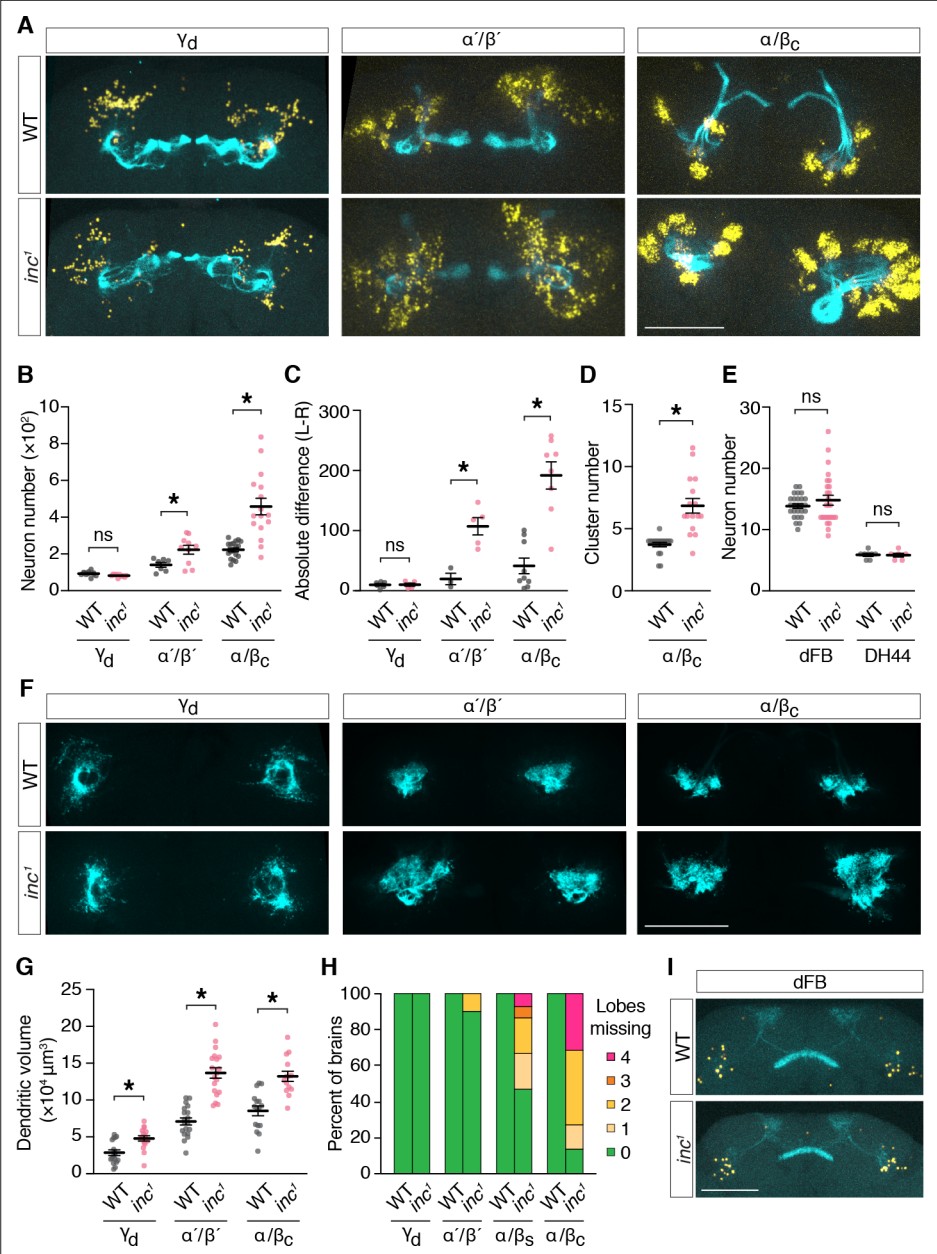

**Figure 7.** *inc* regulates neurogenesis and anatomy of late-born mushroom body (MB) neurons. (**A**) Adult control and *inc¹* brains expressing *UAS-MyrGFP-2A-RedStinger* in indicated MB neuron subtypes, stained with anti-GFP (cyan) and anti-dsRed (yellow). (**B**) MB neuron number per hemisphere. γ_d, *n* = 10–11; α′/β′, *n* = 7–10; α/β_c, *n* = 16–18. *p < 0.01, Welch's *t*-test. (**C**) Absolute difference in MB neuron number between left and right brain hemispheres; γ_d, *n* = 5–6; α′/β′, *n* = 3–5; α/β_c, *n* = 8–9. *p < 0.01, Welch's *t*-test. (**D**) Number of α/β_c neuron clusters per hemisphere. *n* = 16–18. *p < 0.01, Welch's *t*-test. (**E**) Numbers of dorsal fan-shaped body (dFB) and DH44⁺ neurons. dFB, *n* = 26; DH44⁺, *n* = 6–8. ns, p > 0.01, Welch's *t*-test. (**F**) Adult control and *inc¹* brains expressing *UAS-DenMark-smGdP-V5* in indicated MB neuron subtypes, stained with anti-GFP. (**G**) Dendrite volume per hemisphere. γ_d, *n* = 16–17; α′/β′, *n* = 19; α/β_c, *n* = 14–16. *p < 0.01, Welch's *t*-test. (**H**) Quantification of axonal projection defects for MB neuron subtypes. Colored bars represent the number of MB lobes in each brain entirely lacking axonal myr-GFP signal. See also panel (**A**). *n* = 10–25. (**I**) Adult control and *inc¹* brains expressing *UAS-MyrGFP-2A-RedStinger* in dFB neurons. All scale bars represent 100 μm. For (**B–E**) and (**G**), bars represent mean ± SEM.

The online version of this article includes the following figure supplement(s) for figure 7:

**Figure supplement 1.** Analysis of α/β_c neuron clusters and projections.

**Figure supplement 2.** Analysis of dendrites in additional sleep-regulatory circuits.

Our results underscore the importance of unbiased temporal genetic manipulations to define critical periods through which genes impact sleep, and suggest that genes may influence sleep through unappreciated developmental mechanisms. A clear implication of these findings is that variations in human sleep patterns, including pathological disruptions of sleep, may have a developmental origin.

Reciprocal conditional manipulations have been critical in revealing surprising developmental and adult contributions of genes to neuronal function and behavior. In one notable example, anxiety-like behaviors in mice caused by mutations of the 5-HT1A serotonin receptor were found to be rescued by developmental expression of the receptor (*Gross et al., 2002*). Withdrawal of receptor expression in adulthood had no measurable consequences on anxiety-like behavior, and adult-specific receptor expression failed to provide rescue, indicating the necessity and sufficiency of the receptor during development (*Gross et al., 2002*). A second noteworthy example is provided by a mouse model of Rett syndrome, a neurodevelopmental disorder caused by mutation of *MECP2*, a transcriptional regulator. Conditional MeCP2 expression solely in adulthood was found to be sufficient to rescue mutant phenotypes, indicating a critical period for MeCP2 function in adults rather than during brain development (*Guy et al., 2007*; *Guy et al., 2012*). Inactivation of *MECP2* specifically in adulthood causes *MECP2* mutant phenotypes (*McGraw et al., 2011*), confirming its adult requirement. By analogy, various genes that influence sleep might act developmentally or in adulthood in a manner that cannot be anticipated in the absence of conditional manipulations.

*inc* activity is required in neurons for normal sleep, and conversely, restoring *inc* solely to neurons is largely sufficient to rescue the short sleep of *inc* mutants (*Pfeiffenberger and Allada, 2012*; *Stavropoulos and Young, 2011*). Our conditional neuronal manipulations of *inc* span embryonic development through adulthood and indicate that *inc* expression in neurons of late third instar larvae and pupae is sufficient to rescue sleep in *inc* mutants to near wild-type levels, indistinguishable from the rescue provided by constitutive neuronal *inc* expression (*Pfeiffenberger and Allada, 2012*; *Stavropoulos and Young, 2011*). Extending this developmental pulse of neuronal *inc* expression into adulthood does not augment the rescue of *inc* sleep phenotypes, nor does expressing *inc* only in adult neurons restore sleep to *inc* animals. *inc* expression in embryonic, early larval, and adult neurons thus appears dispensable for normal sleep. Instead, *inc* is required at a time coincident with the birth and development of many adult neurons, including those of the MB (*Ito and Hotta, 1992*; *Lee et al., 1999*; *White and Kankel, 1978*). While our findings suggest that the MB is not the sole brain structure through which *inc* impacts sleep, they establish a vital role for *inc* in regulating MB development and its sleep-regulatory functions.

Our findings reveal that *inc* governs neurogenesis, a fundamental process regulated by genes and pathways conserved from flies to mammals (*Doe, 2008*; *Knoblich, 2008*), and suggest that alterations of neurogenesis can cause lasting changes in sleep–wake behavior. The cellular and molecular mechanisms underlying altered neurogenesis in *inc* mutants, including the stochastic nature of these phenotypes and their apparent restriction to the MB, are of particular interest. *inc* null mutations are viable (*Stavropoulos and Young, 2011*), in contrast to the lethality of mutations that globally alter neurogenesis (*Betschinger et al., 2006*; *Lee et al., 2006a*; *Lee et al., 2006b*; *Rolls et al., 2003*; *Vaessin et al., 1991*), consistent with the notion that altered neurogenesis in *inc* mutants manifests preferentially or specifically within the MB. The stochastic nature of neurogenic defects in *inc* mutants and the overproduction of neurons with projection defects are reminiscent of phenotypes of *mushroom body defect* (*mud*) mutants (*Guan et al., 2000*; *Hovhanyan and Raabe, 2009*; *Prokop and Technau, 1994*). In *mud* mutants, infrequent errors in asymmetric neuroblast division give rise to excess neuroblasts and MB neurons (*Bowman et al., 2006*; *Siller et al., 2006*). Similar alterations in neuroblast proliferation in *inc* mutants may account for the stochastic and cumulative defects in the production of late-born MB neurons; a subtle defect in neuroblast proliferation would be expected to manifest particularly in the MB lineage, the longest in the fly brain. Our results do not yet distinguish the cellular populations through which *inc* regulates neurogenesis. One possibility is that *inc* acts in neurons to promote their differentiation, analogous to *lola* and *midlife crisis*, genes whose absence causes neurons to dedifferentiate and acquire the proliferative character of neuroblasts (*Carney et al., 2013*; *Southall et al., 2014*). Another possibility is that *inc* functions in neuroblasts, like *mud*, to govern their asymmetric division.

Our studies and recent findings (*Gong et al., 2021*) suggest that proper regulation of neurogenesis is essential for normal sleep and that altered neurogenesis in discrete circuits can cause lifelong sleep

dysfunction. Intriguing but fragmentary evidence suggests that other genes whose mutation impacts sleep might similarly alter neurogenesis. *wide awake (wake)*, whose mutation causes short sleep in *Drosophila* (*Liu et al., 2014*; *Zhang et al., 2015*), was characterized in an independent study as *banderuola (bnd)* and shown to regulate the asymmetric division of neuroblasts (*Mauri et al., 2014*). An interesting possibility yet to be assessed is whether sleep phenotypes of *wake/bnd* mutants might arise developmentally or through neuroblasts. Similarly, while short sleep phenotypes caused by mutations in the potassium channel subunits encoded by *Shaker* and *Hyperkinetic* (*Bushey et al., 2007*; *Cirelli et al., 2005*) are thought to reflect their role in regulating excitability in specific adult neurons (*Kempf et al., 2019*; *Pimentel et al., 2016*), developmental functions that could contribute to their impact on sleep remain unexplored. Notably, mutations in the Shaker ortholog Kv1.1 analogous to those that strongly reduce sleep in *Drosophila* (*Cirelli et al., 2005*; *Gisselmann et al., 1989*) cause megencephaly and neuronal overproduction in mammals, implicating Kv1.1 in regulating neurogenesis (*Chou et al., 2021*; *Donahue et al., 1996*; *Petersson et al., 2003*; *Yang et al., 2012*). Explicit tests of whether *wake/bnd* and *Shaker* impact sleep through adult or developmental mechanisms, or through a combination of the two, await conditional temporal analysis.

While further manipulations of *inc* are required to elucidate the precise developmental mechanisms by which it impacts sleep, *Cul3* is known to regulate various aspects of neuronal development. Clonal analysis of *Cul3* mutations in *Drosophila* indicates that *Cul3* is required for normal axonal arborization and dendritic elaboration within the MB, as well as axonal fasciculation (*Zhu et al., 2005*). These phenotypes overlap those of *inc* mutants, although direct comparisons are complicated by the pleiotropic nature of *Cul3* mutations, which dysregulate multiple adaptor and substrate pathways. Mosaic analysis of *inc* is required to discern its developmental functions in postmitotic neurons, to compare its phenotypes with *Cul3*, and to distinguish cell autonomous and non-cell autonomous mechanisms. In mammals, *Cul3* mutations alter neurogenesis, cortical lamination, neuronal migration, synaptic development, and cause behavioral deficits (*Amar et al., 2021*; *Dong et al., 2020*; *Fischer et al., 2020*; *Rapanelli et al., 2021*). *inc* and Cul3 are present at synapses in flies and mammals (*Kikuma et al., 2019*; *Li et al., 2017*) and are required at the *Drosophila* larval neuromuscular junction for synaptic homeostasis (*Kikuma et al., 2019*), a process proposed to be a core function of sleep (*Tononi and Cirelli, 2003*). The impact of *inc* on the development and function of central synapses has yet to be assessed, and whether such functions contribute to *inc* sleep phenotypes remains unknown. As a Cul3 adaptor, *inc* may engage multiple molecular targets and cellular pathways. Identifying and manipulating *inc* substrates are thus important goals in elucidating the mechanisms through which *inc* impacts neuronal development and sleep–wake behavior.

The loss of *inc* causes enduring developmental and functional impairments in the MB, a structure important for sensory integration, learning, and sleep regulation. The MB integrates olfactory (*de Belle and Heisenberg, 1994*; *Heisenberg et al., 1985*), gustatory (*Keene and Masek, 2012*; *Masek and Scott, 2010*), visual (*Li et al., 2020*; *Vogt et al., 2016*), and thermal inputs (*Frank et al., 2015*; *Hong et al., 2008*; *Shih et al., 2015*), and its activity is altered by sleep pressure (*Bushey et al., 2015*; *Sitaraman et al., 2015a*). The MB may thus integrate and filter sensory stimuli to promote sleep in appropriate environmental conditions, in a manner modulated by learning and sleep history. The anatomical defects in *inc* mutants may render the MB hypersensitive to sensory stimuli, alter functions of the MB that link learning and sleep (*Berry et al., 2015*; *Cervantes-Sandoval et al., 2017*; *Haynes et al., 2015*; *Seugnet et al., 2011*; *Seugnet et al., 2008*), or impair the relay of sensory input from MB neurons to downstream sleep-promoting circuits (*Aso et al., 2014b*; *Sitaraman et al., 2015a*). While MB circuits and genetic pathways that act in the MB to influence sleep have been manipulated with increasing precision (*Aso et al., 2014b*; *Cavanaugh et al., 2016*; *Guo et al., 2011*; *Joiner et al., 2006*; *Pitman et al., 2006*; *Sitaraman et al., 2015a*; *Sitaraman et al., 2015b*; *Yi et al., 2013*), much remains unknown about the function of the MB in sleep regulation, and additional analysis is required to elucidate how *inc* lesions might alter discrete circuits within the MB and signaling to their targets.

While sensory hypersensitivity and sleep dysfunction are hallmarks of autism and other neurodevelopmental disorders, the underlying mechanisms remain obscure. Given the conserved functions of Cul3–*inc* complexes and the associations of *Cul3* lesions with autism (*Kong et al., 2012*; *Li et al., 2017*; *O'Roak et al., 2012*), elucidating *inc* substrates and their contributions to neurogenesis and neuronal anatomy may provide insights into brain development, tumorigenesis, and sleep disorders.

# Materials and methods

**Key resources table**

| Reagent type (species) or resource | Designation | Source or reference | Identifiers | Additional information |
|---|---|---|---|---|
| Antibody | α-HA (rat monoclonal) | Roche | Cat# 11867431001, RRID:AB_390919 | (1:100) |
| Antibody | α-Brp (mouse monoclonal) | DSHB | Cat# nc82, RRID:AB_2314866 | (1:20 and 1:50) |
| Antibody | α-FLAG (mouse monoclonal) | Sigma-Aldrich | Cat# F1804, RRID:AB_262044 | (1:100) |
| Antibody | α-GFP (mouse monoclonal) | DSHB | Cat# GFP-G1, RRID:AB_2619561 | (1:1000) |
| Antibody | α-GFP (rabbit polyclonal) | Thermo Fisher Scientific | Cat# A11122, RRID:AB_221569 | (1:2000) |
| Antibody | α-dsRed (rabbit polyclonal) | Takara Bio | Cat# 632496, RRID:AB_10013483 | (1:1000) |
| Antibody | α-FasII (mouse monoclonal) | DSHB | Cat# 8 C6, RRID:AB_2314391 | (1:50) |
| Antibody | α-mouse Alexa Fluor 488 (donkey polyclonal) | Thermo Fisher Scientific | Cat# A21202, RRID:AB_141607 | (1:1000) |
| Antibody | α-rabbit Alexa Fluor 488 (donkey polyclonal) | Thermo Fisher Scientific | Cat# A21206, RRID:AB_2535792 | (1:1000) |
| Antibody | α-rat Alexa Fluor 488 (donkey polyclonal) | Thermo Fisher Scientific | Cat# A21208, RRID:AB_2535794 | (1:1000) |
| Antibody | α-rabbit Alexa Fluor 568 (donkey polyclonal) | Thermo Fisher Scientific | Cat# A10042, RRID:AB_2534017 | (1:1000) |
| Antibody | α-mouse Alexa Fluor 647 (donkey polyclonal) | Thermo Fisher Scientific | Cat# A31571, RRID:AB_162542 | (1:1000) |
| Chemical compound, drug | Hydroxyurea | Sigma-Aldrich | H8627 | |
| Genetic reagent (*D. melanogaster*) | $w^{1118}$ | Bloomington *Drosophila* Stock Center | RRID:BDSC_5905 | *Ryder et al., 2004* |
| Genetic reagent (*D. melanogaster*) | $inc^1$ | Stavropoulos lab | FLYB:FBal0266013 | *Stavropoulos and Young, 2011*; BDSC #5,905 background |
| Genetic reagent (*D. melanogaster*) | $inc^2$ | Stavropoulos lab | FLYB:FBal0162225 | *Stavropoulos and Young, 2011*; BDSC #5,905 background |
| Genetic reagent (*D. melanogaster*) | *tub-QS*; *nsyb-Gal4QF* | Christopher Potter | | *Riabinina et al., 2015*; *Li and Stavropoulos, 2016*; BDSC #5,905 background |
| Genetic reagent (*D. melanogaster*) | *inc-Gal4* | Stavropoulos lab | | *Stavropoulos and Young, 2011*; BDSC #5,905 background |
| Genetic reagent (*D. melanogaster*) | $inc^1 inc$-Gal4 | Stavropoulos lab | | *Li et al., 2017*; BDSC #5,905 background |
| Genetic reagent (*D. melanogaster*) | *nsyb-Gal4* | Julie Simpson | | *Simpson, 2016*; BDSC #5,905 background |
| Genetic reagent (*D. melanogaster*) | *c253-Gal4* (MB) | Bloomington *Drosophila* Stock Center | RRID:BDSC_6980 | *Pitman et al., 2006*; BDSC #5,905 background; used in inc[2] rescue screen |
| Genetic reagent (*D. melanogaster*) | *c309-Gal4* (MB) | Bloomington *Drosophila* Stock Center | RRID:BDSC_6906 | *Connolly et al., 1996*; *Pitman et al., 2006 Joiner et al., 2006*; *Aso et al., 2009*; BDSC #5,905 background; used in inc[2] rescue screen |

*Continued on next page*

*Continued*

| Reagent type (species) or resource | Designation | Source or reference | Identifiers | Additional information |
|---|---|---|---|---|
| Genetic reagent (*D. melanogaster*) | c929-Gal4 (l-LNv) | Amita Sehgal | | *Hewes et al., 2000*; *Hewes et al., 2003*; *Sheeba et al., 2008*; *Parisky et al., 2008*; *Shang et al., 2008*; iso31 background; used in inc[2] rescue screen |
| Genetic reagent (*D. melanogaster*) | c584-Gal4 (PI, PPM3) | Amita Sehgal | | *Martin et al., 1999*; *Dubowy et al., 2016*; iso31 background; used in inc[2] rescue screen |
| Genetic reagent (*D. melanogaster*) | R69F08-Gal4 (EB) | Mark Wu | | *Liu et al., 2016*; used in inc[2] rescue screen |
| Genetic reagent (*D. melanogaster*) | R24B11-Gal4 (Helicon) | Bloomington *Drosophila* Stock Center | RRID:BDSC_49070 | *Donlea et al., 2018* |
| Genetic reagent (*D. melanogaster*) | R23E10-Gal4 (dFB) | Bloomington *Drosophila* Stock Center | RRID:BDSC_49032 | *Donlea et al., 2014* |
| Genetic reagent (*D. melanogaster*) | NP2721-Gal4 (DPM) | Leslie Griffith | | *Wu et al., 2011*; *Haynes et al., 2015*; used in inc[2] rescue screen |
| Genetic reagent (*D. melanogaster*) | DH44-Gal4 | Bloomington *Drosophila* Stock Center | RRID:BDSC_39347 | *Cavanaugh et al., 2014* |
| Genetic reagent (*D. melanogaster*) | pdf-Gal4 | Stavropoulos lab | | *Renn et al., 1999* |
| Genetic reagent (*D. melanogaster*) | crz-Gal4 | Stavropoulos lab | | *Tayler et al., 2012* |
| Genetic reagent (*D. melanogaster*) | MB004B (pan-MB) | Yoshinori Aso | | *Sitaraman et al., 2015a* |
| Genetic reagent (*D. melanogaster*) | MB607B (γd) | Yoshinori Aso | | *Sitaraman et al., 2015a* |
| Genetic reagent (*D. melanogaster*) | MB370B (α'β'm + α'β'ap) | Yoshinori Aso | | *Sitaraman et al., 2015a* |
| Genetic reagent (*D. melanogaster*) | MB185B (αβs) | Yoshinori Aso | | *Sitaraman et al., 2015a* |
| Genetic reagent (*D. melanogaster*) | MB594B (αβc) | Yoshinori Aso | | *Sitaraman et al., 2015a* |
| Genetic reagent (*D. melanogaster*) | MB-Gal80 | Michael Young | | *Krashes et al., 2007* |
| Genetic reagent (*D. melanogaster*) | UAS-3xFLAG-Inc | Stavropoulos lab | | *Li et al., 2017*; BDSC #5,905 background |
| Genetic reagent (*D. melanogaster*) | UAS-inc-HA | Stavropoulos lab | | *Li et al., 2017*; BDSC #5,905 background |
| Genetic reagent (*D. melanogaster*) | UAS-inc-RNAi | Vienna *Drosophila* Resource Center | FLYB:FBst0453067 | *Dietzl et al., 2007*; *Stavropoulos and Young, 2011* |
| Genetic reagent (*D. melanogaster*) | UAS-dcr2 | Bloomington *Drosophila* Stock Center | RRID:BDSC_24651 | *Dietzl et al., 2007*; BDSC #5,905 background |
| Genetic reagent (*D. melanogaster*) | UAS-TrpA1 | Stavropoulos lab | | *Hamada et al., 2008*; BDSC #5,905 background |
| Genetic reagent (*D. melanogaster*) | UAS-MyrGFP-2A-RedStinger | Barry Ganetzky | | *Daniels et al., 2014* |

*Continued on next page*

*Continued*

| Reagent type (species) or resource | Designation | Source or reference | Identifiers | Additional information |
|---|---|---|---|---|
| Genetic reagent (*D. melanogaster*) | *5xUAS-DenMark::smGdP-V5* | Bloomington *Drosophila* Stock Center | RRID:BDSC_62138 | ***Nern et al., 2015*** |
| Genetic reagent (*D. melanogaster*) | *5xUAS-IVS-Syt1::smGdP-HA* | Bloomington *Drosophila* Stock Center | RRID:BDSC_62142 | ***Nern et al., 2015*** |
| Genetic reagent (*D. melanogaster*) | *20xUAS-IVS-CD8-GFP* | Bloomington *Drosophila* Stock Center | RRID:BDSC_32194 | ***Pfeiffer et al., 2010*** |
| Genetic reagent (*D. melanogaster*) | *NP1227-Gal4* | Kathy Nagel | | ***Okada et al., 2009***; used in inc[2] rescue screen |
| Genetic reagent (*D. melanogaster*) | *R2-Split Gal4* | Greg Suh | | ***Liu et al., 2016***; used in inc[2] rescue screen |
| Genetic reagent (*D. melanogaster*) | *R72G06-Gal4* | Mark Wu | | used in inc[2] rescue screen |
| Genetic reagent (*D. melanogaster*) | *VT64246-Gal4* | Leslie Griffith | | used in inc[2] rescue screen |
| Genetic reagent (*D. melanogaster*) | *c305a-Gal4* | Leslie Griffith | | used in inc[2] rescue screen |
| Genetic reagent (*D. melanogaster*) | P{GMR49E09-GAL4}attP2 | Bloomington *Drosophila* Stock Center | RRID:BDSC_38692 | ***Jenett et al., 2012***; used in inc[2] rescue screen |
| Genetic reagent (*D. melanogaster*) | P{GMR49F01-GAL4}attP2 | Bloomington *Drosophila* Stock Center | RRID:BDSC_38694 | ***Jenett et al., 2012***; used in inc[2] rescue screen |
| Genetic reagent (*D. melanogaster*) | P{GMR49F02-GAL4}attP2 | Bloomington *Drosophila* Stock Center | RRID:BDSC_38695 | ***Jenett et al., 2012***; used in inc[2] rescue screen |
| Genetic reagent (*D. melanogaster*) | P{GMR49G06-GAL4}attP2 | Bloomington *Drosophila* Stock Center | RRID:BDSC_38707 | ***Jenett et al., 2012***; used in inc[2] rescue screen |
| Genetic reagent (*D. melanogaster*) | P{GMR51G05-GAL4}attP2 | Bloomington *Drosophila* Stock Center | RRID:BDSC_38797 | ***Jenett et al., 2012***; used in inc[2] rescue screen |
| Genetic reagent (*D. melanogaster*) | P{GMR53B06-GAL4}attP2 | Bloomington *Drosophila* Stock Center | RRID:BDSC_38863 | ***Jenett et al., 2012***; used in inc[2] rescue screen |
| Genetic reagent (*D. melanogaster*) | P{GMR53C04-GAL4}attP2 | Bloomington *Drosophila* Stock Center | RRID:BDSC_38871 | ***Jenett et al., 2012***; used in inc[2] rescue screen |
| Genetic reagent (*D. melanogaster*) | P{GMR54F06-GAL4}attP2 | Bloomington *Drosophila* Stock Center | RRID:BDSC_39081 | ***Jenett et al., 2012***; used in inc[2] rescue screen |
| Genetic reagent (*D. melanogaster*) | P{GMR55A03-GAL4}attP2 | Bloomington *Drosophila* Stock Center | RRID:BDSC_39095 | ***Jenett et al., 2012***; used in inc[2] rescue screen |
| Genetic reagent (*D. melanogaster*) | P{GMR55B12-GAL4}attP2 | Bloomington *Drosophila* Stock Center | RRID:BDSC_39103 | ***Jenett et al., 2012***; used in inc[2] rescue screen |
| Genetic reagent (*D. melanogaster*) | P{GMR55D01-GAL4}attP2 | Bloomington *Drosophila* Stock Center | RRID:BDSC_39110 | ***Jenett et al., 2012***; used in inc[2] rescue screen |

*Continued on next page*

*Continued*

| Reagent type (species) or resource | Designation | Source or reference | Identifiers | Additional information |
|---|---|---|---|---|
| Genetic reagent (*D. melanogaster*) | P{GMR55D05-GAL4}attP2 | Bloomington *Drosophila* Stock Center | RRID:BDSC_39112 | *Jenett et al., 2012*; used in inc[2] rescue screen |
| Genetic reagent (*D. melanogaster*) | P{GMR55F07-GAL4}attP2 | Bloomington *Drosophila* Stock Center | RRID:BDSC_39128 | *Jenett et al., 2012*; used in inc[2] rescue screen |
| Genetic reagent (*D. melanogaster*) | P{GMR55G11-GAL4}attP2 | Bloomington *Drosophila* Stock Center | RRID:BDSC_39132 | *Jenett et al., 2012*; used in inc[2] rescue screen |
| Genetic reagent (*D. melanogaster*) | P{GMR56H02-GAL4}attP2 | Bloomington *Drosophila* Stock Center | RRID:BDSC_39164 | *Jenett et al., 2012*; used in inc[2] rescue screen |
| Genetic reagent (*D. melanogaster*) | P{GMR56H09-GAL4}attP2 | Bloomington *Drosophila* Stock Center | RRID:BDSC_39166 | *Jenett et al., 2012*; used in inc[2] rescue screen |
| Genetic reagent (*D. melanogaster*) | P{GMR58E10-GAL4}attP2 | Bloomington *Drosophila* Stock Center | RRID:BDSC_39184 | *Jenett et al., 2012*; used in inc[2] rescue screen |
| Genetic reagent (*D. melanogaster*) | P{GMR58H05-GAL4}attP2 | Bloomington *Drosophila* Stock Center | RRID:BDSC_39198 | *Jenett et al., 2012*; used in inc[2] rescue screen |
| Genetic reagent (*D. melanogaster*) | P{GMR59B10-GAL4}attP2 | Bloomington *Drosophila* Stock Center | RRID:BDSC_39209 | *Jenett et al., 2012*; used in inc[2] rescue screen |
| Genetic reagent (*D. melanogaster*) | P{GMR59E09-GAL4}attP2 | Bloomington *Drosophila* Stock Center | RRID:BDSC_39220 | *Jenett et al., 2012*; used in inc[2] rescue screen |
| Genetic reagent (*D. melanogaster*) | P{GMR59H05-GAL4}attP2 | Bloomington *Drosophila* Stock Center | RRID:BDSC_39229 | *Jenett et al., 2012*; used in inc[2] rescue screen |
| Genetic reagent (*D. melanogaster*) | P{GMR60C01-GAL4}attP2 | Bloomington *Drosophila* Stock Center | RRID:BDSC_39240 | *Jenett et al., 2012*; used in inc[2] rescue screen |
| Genetic reagent (*D. melanogaster*) | P{GMR60D05-GAL4}attP2 | Bloomington *Drosophila* Stock Center | RRID:BDSC_39247 | *Jenett et al., 2012*; used in inc[2] rescue screen |
| Genetic reagent (*D. melanogaster*) | P{GMR60H12-GAL4}attP2 | Bloomington *Drosophila* Stock Center | RRID:BDSC_39268 | *Jenett et al., 2012*; used in inc[2] rescue screen |
| Genetic reagent (*D. melanogaster*) | P{GMR64A11-GAL4}attP2 | Bloomington *Drosophila* Stock Center | RRID:BDSC_39289 | *Jenett et al., 2012*; used in inc[2] rescue screen |
| Genetic reagent (*D. melanogaster*) | P{GMR64F03-GAL4}attP2 | Bloomington *Drosophila* Stock Center | RRID:BDSC_39309 | *Jenett et al., 2012*; used in inc[2] rescue screen |
| Genetic reagent (*D. melanogaster*) | P{GMR64G05-GAL4}attP2 | Bloomington *Drosophila* Stock Center | RRID:BDSC_39316 | *Jenett et al., 2012*; used in inc[2] rescue screen |
| Genetic reagent (*D. melanogaster*) | P{GMR65B04-GAL4}attP2 | Bloomington *Drosophila* Stock Center | RRID:BDSC_39336 | *Jenett et al., 2012*; used in inc[2] rescue screen |

*Continued*

| Reagent type (species) or resource | Designation | Source or reference | Identifiers | Additional information |
|---|---|---|---|---|
| Genetic reagent (*D. melanogaster*) | P{GMR65D06-GAL4}attP2 | Bloomington *Drosophila* Stock Center | RRID:BDSC_39352 | *Jenett et al., 2012*; used in inc[2] rescue screen |
| Genetic reagent (*D. melanogaster*) | P{GMR65D07-GAL4}attP2 | Bloomington *Drosophila* Stock Center | RRID:BDSC_39353 | *Jenett et al., 2012*; used in inc[2] rescue screen |
| Genetic reagent (*D. melanogaster*) | P{GMR67A04-GAL4}attP2 | Bloomington *Drosophila* Stock Center | RRID:BDSC_39396 | *Jenett et al., 2012*; used in inc[2] rescue screen |
| Genetic reagent (*D. melanogaster*) | P{GMR69C02-GAL4}attP2 | Bloomington *Drosophila* Stock Center | RRID:BDSC_39483 | *Jenett et al., 2012*; used in inc[2] rescue screen |
| Genetic reagent (*D. melanogaster*) | P{GMR71D01-GAL4}attP2 | Bloomington *Drosophila* Stock Center | RRID:BDSC_39579 | *Jenett et al., 2012*; used in inc[2] rescue screen |
| Genetic reagent (*D. melanogaster*) | P{GMR72H03-GAL4}attP2 | Bloomington *Drosophila* Stock Center | RRID:BDSC_39799 | *Jenett et al., 2012*; used in inc[2] rescue screen |
| Genetic reagent (*D. melanogaster*) | P{GMR74H01-GAL4}attP2 | Bloomington *Drosophila* Stock Center | RRID:BDSC_39872 | *Jenett et al., 2012*; used in inc[2] rescue screen |
| Genetic reagent (*D. melanogaster*) | P{GMR76F06-GAL4}attP2 | Bloomington *Drosophila* Stock Center | RRID:BDSC_39937 | *Jenett et al., 2012*; used in inc[2] rescue screen |
| Genetic reagent (*D. melanogaster*) | P{GMR77H03-GAL4}attP2 | Bloomington *Drosophila* Stock Center | RRID:BDSC_39976 | *Jenett et al., 2012*; used in inc[2] rescue screen |
| Genetic reagent (*D. melanogaster*) | P{GMR78A01-GAL4}attP2 | Bloomington *Drosophila* Stock Center | RRID:BDSC_39985 | *Jenett et al., 2012*; used in inc[2] rescue screen |
| Genetic reagent (*D. melanogaster*) | P{GMR78G06-GAL4}attP2 | Bloomington *Drosophila* Stock Center | RRID:BDSC_40013 | *Jenett et al., 2012*; used in inc[2] rescue screen |
| Genetic reagent (*D. melanogaster*) | P{GMR79A01-GAL4}attP2 | Bloomington *Drosophila* Stock Center | RRID:BDSC_40021 | *Jenett et al., 2012*; used in inc[2] rescue screen |
| Genetic reagent (*D. melanogaster*) | P{GMR79B08-GAL4}attP2 | Bloomington *Drosophila* Stock Center | RRID:BDSC_40029 | *Jenett et al., 2012*; used in inc[2] rescue screen |
| Genetic reagent (*D. melanogaster*) | P{GMR83H01-GAL4}attP2 | Bloomington *Drosophila* Stock Center | RRID:BDSC_40368 | *Jenett et al., 2012*; used in inc[2] rescue screen |
| Genetic reagent (*D. melanogaster*) | P{GMR85C07-GAL4}attP2 | Bloomington *Drosophila* Stock Center | RRID:BDSC_40422 | *Jenett et al., 2012*; used in inc[2] rescue screen |
| Genetic reagent (*D. melanogaster*) | P{GMR87A08-GAL4}attP2 | Bloomington *Drosophila* Stock Center | RRID:BDSC_40473 | *Jenett et al., 2012*; used in inc[2] rescue screen |
| Genetic reagent (*D. melanogaster*) | P{GMR92G09-GAL4}attP2 | Bloomington *Drosophila* Stock Center | RRID:BDSC_40629 | *Jenett et al., 2012*; used in inc[2] rescue screen |

*Continued*

| Reagent type (species) or resource | Designation | Source or reference | Identifiers | Additional information |
|---|---|---|---|---|
| Genetic reagent (*D. melanogaster*) | P{GMR93C06-GAL4}attP2 | Bloomington *Drosophila* Stock Center | RRID:BDSC_40647 | *Jenett et al., 2012*; used in inc[2] rescue screen |
| Genetic reagent (*D. melanogaster*) | P{GMR93G05-GAL4}attP2 | Bloomington *Drosophila* Stock Center | RRID:BDSC_40662 | *Jenett et al., 2012*; used in inc[2] rescue screen |
| Genetic reagent (*D. melanogaster*) | P{GMR93H07-GAL4}attP2 | Bloomington *Drosophila* Stock Center | RRID:BDSC_40669 | *Jenett et al., 2012*; used in inc[2] rescue screen |
| Genetic reagent (*D. melanogaster*) | P{GMR94D04-GAL4}attP2 | Bloomington *Drosophila* Stock Center | RRID:BDSC_40681 | *Jenett et al., 2012*; used in inc[2] rescue screen |
| Genetic reagent (*D. melanogaster*) | P{GMR94E07-GAL4}attP2 | Bloomington *Drosophila* Stock Center | RRID:BDSC_40688 | *Jenett et al., 2012*; used in inc[2] rescue screen |
| Genetic reagent (*D. melanogaster*) | P{GMR94F06-GAL4}attP2 | Bloomington *Drosophila* Stock Center | RRID:BDSC_40694 | *Jenett et al., 2012*; used in inc[2] rescue screen |
| Genetic reagent (*D. melanogaster*) | P{GMR95E08-GAL4}attP2 | Bloomington *Drosophila* Stock Center | RRID:BDSC_40710 | *Jenett et al., 2012*; used in inc[2] rescue screen |
| Genetic reagent (*D. melanogaster*) | P{GMR95F11-GAL4}attP2 | Bloomington *Drosophila* Stock Center | RRID:BDSC_40714 | *Jenett et al., 2012*; used in inc[2] rescue screen |
| Genetic reagent (*D. melanogaster*) | P{GMR40B09-GAL4}attP2 | Bloomington *Drosophila* Stock Center | RRID:BDSC_41235 | *Jenett et al., 2012*; used in inc[2] rescue screen |
| Genetic reagent (*D. melanogaster*) | P{GMR40E08-GAL4}attP2 | Bloomington *Drosophila* Stock Center | RRID:BDSC_41238 | *Jenett et al., 2012*; used in inc[2] rescue screen |
| Genetic reagent (*D. melanogaster*) | P{GMR41G11-GAL4}attP2 | Bloomington *Drosophila* Stock Center | RRID:BDSC_41244 | *Jenett et al., 2012*; used in inc[2] rescue screen |
| Genetic reagent (*D. melanogaster*) | P{GMR42F06-GAL4}attP2 | Bloomington *Drosophila* Stock Center | RRID:BDSC_41253 | *Jenett et al., 2012*; used in inc[2] rescue screen |
| Genetic reagent (*D. melanogaster*) | P{GMR60D10-GAL4}attP2 | Bloomington *Drosophila* Stock Center | RRID:BDSC_41284 | *Jenett et al., 2012*; used in inc[2] rescue screen |
| Genetic reagent (*D. melanogaster*) | P{GMR65C03-GAL4}attP2 | Bloomington *Drosophila* Stock Center | RRID:BDSC_41290 | *Jenett et al., 2012*; used in inc[2] rescue screen |
| Genetic reagent (*D. melanogaster*) | P{GMR74B11-GAL4}attP2 | Bloomington *Drosophila* Stock Center | RRID:BDSC_41301 | *Jenett et al., 2012*; used in inc[2] rescue screen |
| Genetic reagent (*D. melanogaster*) | P{GMR87B02-GAL4}attP2 | Bloomington *Drosophila* Stock Center | RRID:BDSC_41316 | *Jenett et al., 2012*; used in inc[2] rescue screen |
| Genetic reagent (*D. melanogaster*) | P{GMR65B09-GAL4}attP2 | Bloomington *Drosophila* Stock Center | RRID:BDSC_41353 | *Jenett et al., 2012*; used in inc[2] rescue screen |

*Continued*

| Reagent type (species) or resource | Designation | Source or reference | Identifiers | Additional information |
|---|---|---|---|---|
| Genetic reagent (*D. melanogaster*) | P{GMR34C12-GAL4}attP2 | Bloomington *Drosophila* Stock Center | RRID:BDSC_45219 | *Jenett et al., 2012*; used in inc[2] rescue screen |
| Genetic reagent (*D. melanogaster*) | P{GMR45D10-GAL4}attP2 | Bloomington *Drosophila* Stock Center | RRID:BDSC_45323 | *Jenett et al., 2012*; used in inc[2] rescue screen |
| Genetic reagent (*D. melanogaster*) | P{GMR60G12-GAL4}attP2 | Bloomington *Drosophila* Stock Center | RRID:BDSC_45360 | *Jenett et al., 2012*; used in inc[2] rescue screen |
| Genetic reagent (*D. melanogaster*) | P{GMR23G07-GAL4}attP2 | Bloomington *Drosophila* Stock Center | RRID:BDSC_45493 | *Jenett et al., 2012*; used in inc[2] rescue screen |
| Genetic reagent (*D. melanogaster*) | P{GMR26C01-GAL4}attP2 | Bloomington *Drosophila* Stock Center | RRID:BDSC_45518 | *Jenett et al., 2012*; used in inc[2] rescue screen |
| Genetic reagent (*D. melanogaster*) | P{GMR48D06-GAL4}attP2 | Bloomington *Drosophila* Stock Center | RRID:BDSC_45774 | *Jenett et al., 2012*; used in inc[2] rescue screen |
| Genetic reagent (*D. melanogaster*) | P{GMR20E01-GAL4}attP2 | Bloomington *Drosophila* Stock Center | RRID:BDSC_45837 | *Jenett et al., 2012*; used in inc[2] rescue screen |
| Genetic reagent (*D. melanogaster*) | P{GMR25G01-GAL4}attP2 | Bloomington *Drosophila* Stock Center | RRID:BDSC_45851 | *Jenett et al., 2012*; used in inc[2] rescue screen |
| Genetic reagent (*D. melanogaster*) | P{GMR53G07-GAL4}attP2 | Bloomington *Drosophila* Stock Center | RRID:BDSC_46041 | *Jenett et al., 2012*; used in inc[2] rescue screen |
| Genetic reagent (*D. melanogaster*) | P{GMR55G02-GAL4}attP2 | Bloomington *Drosophila* Stock Center | RRID:BDSC_46070 | *Jenett et al., 2012*; used in inc[2] rescue screen |
| Genetic reagent (*D. melanogaster*) | P{GMR35H03-GAL4}attP2 | Bloomington *Drosophila* Stock Center | RRID:BDSC_46205 | *Jenett et al., 2012*; used in inc[2] rescue screen |
| Genetic reagent (*D. melanogaster*) | P{GMR46H09-GAL4}attP2 | Bloomington *Drosophila* Stock Center | RRID:BDSC_46275 | *Jenett et al., 2012*; used in inc[2] rescue screen |
| Genetic reagent (*D. melanogaster*) | P{GMR58G05-GAL4}attP2 | Bloomington *Drosophila* Stock Center | RRID:BDSC_46410 | *Jenett et al., 2012*; used in inc[2] rescue screen |
| Genetic reagent (*D. melanogaster*) | P{GMR59H01-GAL4}attP2 | Bloomington *Drosophila* Stock Center | RRID:BDSC_46423 | *Jenett et al., 2012*; used in inc[2] rescue screen |
| Genetic reagent (*D. melanogaster*) | P{GMR64D08-GAL4}attP2 | Bloomington *Drosophila* Stock Center | RRID:BDSC_46539 | *Jenett et al., 2012*; used in inc[2] rescue screen |
| Genetic reagent (*D. melanogaster*) | P{GMR65C05-GAL4}attP2 | Bloomington *Drosophila* Stock Center | RRID:BDSC_46554 | *Jenett et al., 2012*; used in inc[2] rescue screen |
| Genetic reagent (*D. melanogaster*) | P{GMR65H08-GAL4}attP2 | Bloomington *Drosophila* Stock Center | RRID:BDSC_46566 | *Jenett et al., 2012*; used in inc[2] rescue screen |

*Continued*

| Reagent type (species) or resource | Designation | Source or reference | Identifiers | Additional information |
| --- | --- | --- | --- | --- |
| Genetic reagent (*D. melanogaster*) | P{GMR69H02-GAL4}attP2 | Bloomington *Drosophila* Stock Center | RRID:BDSC_46620 | *Jenett et al., 2012*; used in inc[2] rescue screen |
| Genetic reagent (*D. melanogaster*) | P{GMR70G11-GAL4}attP2 | Bloomington *Drosophila* Stock Center | RRID:BDSC_46641 | *Jenett et al., 2012*; used in inc[2] rescue screen |
| Genetic reagent (*D. melanogaster*) | P{GMR71E04-GAL4}attP2 | Bloomington *Drosophila* Stock Center | RRID:BDSC_46658 | *Jenett et al., 2012*; used in inc[2] rescue screen |
| Genetic reagent (*D. melanogaster*) | P{GMR72A04-GAL4}attP2 | Bloomington *Drosophila* Stock Center | RRID:BDSC_46665 | *Jenett et al., 2012*; used in inc[2] rescue screen |
| Genetic reagent (*D. melanogaster*) | P{GMR73D06-GAL4}attP2 | Bloomington *Drosophila* Stock Center | RRID:BDSC_46692 | *Jenett et al., 2012*; used in inc[2] rescue screen |
| Genetic reagent (*D. melanogaster*) | P{GMR56F05-GAL4}attP2 | Bloomington *Drosophila* Stock Center | RRID:BDSC_46714 | *Jenett et al., 2012*; used in inc[2] rescue screen |
| Genetic reagent (*D. melanogaster*) | P{GMR77A04-GAL4}attP2 | Bloomington *Drosophila* Stock Center | RRID:BDSC_46976 | *Jenett et al., 2012*; used in inc[2] rescue screen |
| Genetic reagent (*D. melanogaster*) | P{GMR80C12-GAL4}attP2 | Bloomington *Drosophila* Stock Center | RRID:BDSC_47059 | *Jenett et al., 2012*; used in inc[2] rescue screen |
| Genetic reagent (*D. melanogaster*) | P{GMR81C04-GAL4}attP2 | Bloomington *Drosophila* Stock Center | RRID:BDSC_47087 | *Jenett et al., 2012*; used in inc[2] rescue screen |
| Genetic reagent (*D. melanogaster*) | P{GMR81D04-GAL4}attP2 | Bloomington *Drosophila* Stock Center | RRID:BDSC_47094 | *Jenett et al., 2012*; used in inc[2] rescue screen |
| Genetic reagent (*D. melanogaster*) | P{GMR91A08-GAL4}attP2 | Bloomington *Drosophila* Stock Center | RRID:BDSC_47148 | *Jenett et al., 2012*; used in inc[2] rescue screen |
| Genetic reagent (*D. melanogaster*) | P{GMR91G01-GAL4}attP2 | Bloomington *Drosophila* Stock Center | RRID:BDSC_47175 | *Jenett et al., 2012*; used in inc[2] rescue screen |
| Genetic reagent (*D. melanogaster*) | P{GMR92H11-GAL4}attP2 | Bloomington *Drosophila* Stock Center | RRID:BDSC_47211 | *Jenett et al., 2012*; used in inc[2] rescue screen |
| Genetic reagent (*D. melanogaster*) | P{GMR93B04-GAL4}attP2 | Bloomington *Drosophila* Stock Center | RRID:BDSC_47215 | *Jenett et al., 2012*; used in inc[2] rescue screen |
| Genetic reagent (*D. melanogaster*) | P{GMR93D01-GAL4}attP2 | Bloomington *Drosophila* Stock Center | RRID:BDSC_47221 | *Jenett et al., 2012*; used in inc[2] rescue screen |
| Genetic reagent (*D. melanogaster*) | P{GMR93D06-GAL4}attP2 | Bloomington *Drosophila* Stock Center | RRID:BDSC_47224 | *Jenett et al., 2012*; used in inc[2] rescue screen |
| Genetic reagent (*D. melanogaster*) | P{GMR93G11-GAL4}attP2 | Bloomington *Drosophila* Stock Center | RRID:BDSC_47238 | *Jenett et al., 2012*; used in inc[2] rescue screen |

*Continued*

| Reagent type (species) or resource | Designation | Source or reference | Identifiers | Additional information |
|---|---|---|---|---|
| Genetic reagent (*D. melanogaster*) | P{GMR94H10-GAL4}attP2 | Bloomington *Drosophila* Stock Center | RRID:BDSC_47268 | *Jenett et al., 2012*; used in inc[2] rescue screen |
| Genetic reagent (*D. melanogaster*) | P{GMR16D12-GAL4}attP2 | Bloomington *Drosophila* Stock Center | RRID:BDSC_47325 | *Jenett et al., 2012*; used in inc[2] rescue screen |
| Genetic reagent (*D. melanogaster*) | P{GMR16H05-GAL4}attP2 | Bloomington *Drosophila* Stock Center | RRID:BDSC_47327 | *Jenett et al., 2012*; used in inc[2] rescue screen |
| Genetic reagent (*D. melanogaster*) | P{GMR10E03-GAL4}attP2 | Bloomington *Drosophila* Stock Center | RRID:BDSC_47447 | *Jenett et al., 2012*; used in inc[2] rescue screen |
| Genetic reagent (*D. melanogaster*) | P{GMR42E09-GAL4}attP2 | Bloomington *Drosophila* Stock Center | RRID:BDSC_47589 | *Jenett et al., 2012*; used in inc[2] rescue screen |
| Genetic reagent (*D. melanogaster*) | P{GMR52A01-GAL4}attP2 | Bloomington *Drosophila* Stock Center | RRID:BDSC_47634 | *Jenett et al., 2012*; used in inc[2] rescue screen |
| Genetic reagent (*D. melanogaster*) | P{GMR70A09-GAL4}attP2 | Bloomington *Drosophila* Stock Center | RRID:BDSC_47720 | *Jenett et al., 2012*; used in inc[2] rescue screen |
| Genetic reagent (*D. melanogaster*) | P{GMR72F10-GAL4}attP2 | Bloomington *Drosophila* Stock Center | RRID:BDSC_47731 | *Jenett et al., 2012*; used in inc[2] rescue screen |
| Genetic reagent (*D. melanogaster*) | P{GMR74G04-GAL4}attP2 | Bloomington *Drosophila* Stock Center | RRID:BDSC_47742 | *Jenett et al., 2012*; used in inc[2] rescue screen |
| Genetic reagent (*D. melanogaster*) | P{GMR10A11-GAL4}attP2 | Bloomington *Drosophila* Stock Center | RRID:BDSC_47839 | *Jenett et al., 2012*; used in inc[2] rescue screen |
| Genetic reagent (*D. melanogaster*) | P{GMR10A12-GAL4}attP2 | Bloomington *Drosophila* Stock Center | RRID:BDSC_47840 | *Jenett et al., 2012*; used in inc[2] rescue screen |
| Genetic reagent (*D. melanogaster*) | P{GMR13C06-GAL4}attP2 | Bloomington *Drosophila* Stock Center | RRID:BDSC_47860 | *Jenett et al., 2012*; used in inc[2] rescue screen |
| Genetic reagent (*D. melanogaster*) | P{GMR19G10-GAL4}attP2 | Bloomington *Drosophila* Stock Center | RRID:BDSC_47887 | *Jenett et al., 2012*; used in inc[2] rescue screen |
| Genetic reagent (*D. melanogaster*) | P{GMR21C11-GAL4}attP2 | Bloomington *Drosophila* Stock Center | RRID:BDSC_47898 | *Jenett et al., 2012*; used in inc[2] rescue screen |
| Genetic reagent (*D. melanogaster*) | P{GMR30F07-GAL4}attP2 | Bloomington *Drosophila* Stock Center | RRID:BDSC_47911 | *Jenett et al., 2012*; used in inc[2] rescue screen |
| Genetic reagent (*D. melanogaster*) | P{GMR44G12-GAL4}attP2 | Bloomington *Drosophila* Stock Center | RRID:BDSC_47933 | *Jenett et al., 2012*; used in inc[2] rescue screen |
| Genetic reagent (*D. melanogaster*) | P{GMR52F09-GAL4}attP2 | Bloomington *Drosophila* Stock Center | RRID:BDSC_47943 | *Jenett et al., 2012*; used in inc[2] rescue screen |

*Continued*

| Reagent type (species) or resource | Designation | Source or reference | Identifiers | Additional information |
|---|---|---|---|---|
| Genetic reagent (*D. melanogaster*) | P{GMR28F06-GAL4}attP2 | Bloomington *Drosophila* Stock Center | RRID:BDSC_48083 | *Jenett et al., 2012*; used in inc[2] rescue screen |
| Genetic reagent (*D. melanogaster*) | P{GMR33H11-GAL4}attP2 | Bloomington *Drosophila* Stock Center | RRID:BDSC_48119 | *Jenett et al., 2012*; used in inc[2] rescue screen |
| Genetic reagent (*D. melanogaster*) | P{GMR50A07-GAL4}attP2 | Bloomington *Drosophila* Stock Center | RRID:BDSC_48179 | *Jenett et al., 2012*; used in inc[2] rescue screen |
| Genetic reagent (*D. melanogaster*) | P{GMR51B08-GAL4}attP2 | Bloomington *Drosophila* Stock Center | RRID:BDSC_48183 | *Jenett et al., 2012*; used in inc[2] rescue screen |
| Genetic reagent (*D. melanogaster*) | P{GMR52C05-GAL4}attP2 | Bloomington *Drosophila* Stock Center | RRID:BDSC_48190 | *Jenett et al., 2012*; used in inc[2] rescue screen |
| Genetic reagent (*D. melanogaster*) | P{GMR54H12-GAL4}attP2 | Bloomington *Drosophila* Stock Center | RRID:BDSC_48205 | *Jenett et al., 2012*; used in inc[2] rescue screen |
| Genetic reagent (*D. melanogaster*) | P{GMR58F01-GAL4}attP2 | Bloomington *Drosophila* Stock Center | RRID:BDSC_48213 | *Jenett et al., 2012*; used in inc[2] rescue screen |
| Genetic reagent (*D. melanogaster*) | P{GMR59B11-GAL4}attP2 | Bloomington *Drosophila* Stock Center | RRID:BDSC_48215 | *Jenett et al., 2012*; used in inc[2] rescue screen |
| Genetic reagent (*D. melanogaster*) | P{GMR59C12-GAL4}attP2 | Bloomington *Drosophila* Stock Center | RRID:BDSC_48219 | *Jenett et al., 2012*; used in inc[2] rescue screen |
| Genetic reagent (*D. melanogaster*) | P{GMR59E04-GAL4}attP2 | Bloomington *Drosophila* Stock Center | RRID:BDSC_48221 | *Jenett et al., 2012*; used in inc[2] rescue screen |
| Genetic reagent (*D. melanogaster*) | P{GMR10D10-GAL4}attP2 | Bloomington *Drosophila* Stock Center | RRID:BDSC_48261 | *Jenett et al., 2012*; used in inc[2] rescue screen |
| Genetic reagent (*D. melanogaster*) | P{GMR10H09-GAL4}attP2 | Bloomington *Drosophila* Stock Center | RRID:BDSC_48277 | *Jenett et al., 2012*; used in inc[2] rescue screen |
| Genetic reagent (*D. melanogaster*) | P{GMR67B06-GAL4}attP2 | Bloomington *Drosophila* Stock Center | RRID:BDSC_48294 | *Jenett et al., 2012*; used in inc[2] rescue screen |
| Genetic reagent (*D. melanogaster*) | P{GMR73H09-GAL4}attP2 | Bloomington *Drosophila* Stock Center | RRID:BDSC_48318 | *Jenett et al., 2012*; used in inc[2] rescue screen |
| Genetic reagent (*D. melanogaster*) | P{GMR87C01-GAL4}attP2 | Bloomington *Drosophila* Stock Center | RRID:BDSC_48389 | *Jenett et al., 2012*; used in inc[2] rescue screen |
| Genetic reagent (*D. melanogaster*) | P{GMR89C02-GAL4}attP2 | Bloomington *Drosophila* Stock Center | RRID:BDSC_48404 | *Jenett et al., 2012*; used in inc[2] rescue screen |
| Genetic reagent (*D. melanogaster*) | P{GMR92A08-GAL4}attP2 | Bloomington *Drosophila* Stock Center | RRID:BDSC_48414 | *Jenett et al., 2012*; used in inc[2] rescue screen |

*Continued*

| Reagent type (species) or resource | Designation | Source or reference | Identifiers | Additional information |
|---|---|---|---|---|
| Genetic reagent (*D. melanogaster*) | P{GMR93C08-GAL4}attP2 | Bloomington *Drosophila* Stock Center | RRID:BDSC_48417 | *Jenett et al., 2012*; used in inc[2] rescue screen |
| Genetic reagent (*D. melanogaster*) | P{GMR93F02-GAL4}attP2 | Bloomington *Drosophila* Stock Center | RRID:BDSC_48422 | *Jenett et al., 2012*; used in inc[2] rescue screen |
| Genetic reagent (*D. melanogaster*) | P{GMR95F03-GAL4}attP2 | Bloomington *Drosophila* Stock Center | RRID:BDSC_48433 | *Jenett et al., 2012*; used in inc[2] rescue screen |
| Genetic reagent (*D. melanogaster*) | P{GMR10E07-GAL4}attP2 | Bloomington *Drosophila* Stock Center | RRID:BDSC_48440 | *Jenett et al., 2012*; used in inc[2] rescue screen |
| Genetic reagent (*D. melanogaster*) | P{GMR11C07-GAL4}attP2 | Bloomington *Drosophila* Stock Center | RRID:BDSC_48448 | *Jenett et al., 2012*; used in inc[2] rescue screen |
| Genetic reagent (*D. melanogaster*) | P{GMR12B10-GAL4}attP2 | Bloomington *Drosophila* Stock Center | RRID:BDSC_48490 | *Jenett et al., 2012*; used in inc[2] rescue screen |
| Genetic reagent (*D. melanogaster*) | P{GMR12D12-GAL4}attP2 | Bloomington *Drosophila* Stock Center | RRID:BDSC_48506 | *Jenett et al., 2012*; used in inc[2] rescue screen |
| Genetic reagent (*D. melanogaster*) | P{GMR12G09-GAL4}attP2 | Bloomington *Drosophila* Stock Center | RRID:BDSC_48525 | *Jenett et al., 2012*; used in inc[2] rescue screen |
| Genetic reagent (*D. melanogaster*) | P{GMR13B10-GAL4}attP2 | Bloomington *Drosophila* Stock Center | RRID:BDSC_48548 | *Jenett et al., 2012*; used in inc[2] rescue screen |
| Genetic reagent (*D. melanogaster*) | P{GMR13D09-GAL4}attP2 | Bloomington *Drosophila* Stock Center | RRID:BDSC_48561 | *Jenett et al., 2012*; used in inc[2] rescue screen |
| Genetic reagent (*D. melanogaster*) | P{GMR13E04-GAL4}attP2 | Bloomington *Drosophila* Stock Center | RRID:BDSC_48565 | *Jenett et al., 2012*; used in inc[2] rescue screen |
| Genetic reagent (*D. melanogaster*) | P{GMR13E06-GAL4}attP2 | Bloomington *Drosophila* Stock Center | RRID:BDSC_48566 | *Jenett et al., 2012*; used in inc[2] rescue screen |
| Genetic reagent (*D. melanogaster*) | P{GMR13F04-GAL4}attP2 | Bloomington *Drosophila* Stock Center | RRID:BDSC_48573 | *Jenett et al., 2012*; used in inc[2] rescue screen |
| Genetic reagent (*D. melanogaster*) | P{GMR14C08-GAL4}attP2 | Bloomington *Drosophila* Stock Center | RRID:BDSC_48606 | *Jenett et al., 2012*; used in inc[2] rescue screen |
| Genetic reagent (*D. melanogaster*) | P{GMR20F01-GAL4}attP2 | Bloomington *Drosophila* Stock Center | RRID:BDSC_48610 | *Jenett et al., 2012*; used in inc[2] rescue screen |
| Genetic reagent (*D. melanogaster*) | P{GMR14E05-GAL4}attP2 | Bloomington *Drosophila* Stock Center | RRID:BDSC_48642 | *Jenett et al., 2012*; used in inc[2] rescue screen |
| Genetic reagent (*D. melanogaster*) | P{GMR14E06-GAL4}attP2 | Bloomington *Drosophila* Stock Center | RRID:BDSC_48643 | *Jenett et al., 2012*; used in inc[2] rescue screen |

*Continued*

| Reagent type (species) or resource | Designation | Source or reference | Identifiers | Additional information |
|---|---|---|---|---|
| Genetic reagent (*D. melanogaster*) | P{GMR14E09-GAL4}attP2 | Bloomington *Drosophila* Stock Center | RRID:BDSC_48645 | *Jenett et al., 2012*; used in inc[2] rescue screen |
| Genetic reagent (*D. melanogaster*) | P{GMR14E12-GAL4}attP2 | Bloomington *Drosophila* Stock Center | RRID:BDSC_48647 | *Jenett et al., 2012*; used in inc[2] rescue screen |
| Genetic reagent (*D. melanogaster*) | P{GMR14F11-GAL4}attP2 | Bloomington *Drosophila* Stock Center | RRID:BDSC_48653 | *Jenett et al., 2012*; used in inc[2] rescue screen |
| Genetic reagent (*D. melanogaster*) | P{GMR14G08-GAL4}attP2 | Bloomington *Drosophila* Stock Center | RRID:BDSC_48661 | *Jenett et al., 2012*; used in inc[2] rescue screen |
| Genetic reagent (*D. melanogaster*) | P{GMR14H02-GAL4}attP2 | Bloomington *Drosophila* Stock Center | RRID:BDSC_48664 | *Jenett et al., 2012*; used in inc[2] rescue screen |
| Genetic reagent (*D. melanogaster*) | P{GMR15B07-GAL4}attP2 | Bloomington *Drosophila* Stock Center | RRID:BDSC_48678 | *Jenett et al., 2012*; used in inc[2] rescue screen |
| Genetic reagent (*D. melanogaster*) | P{GMR15D11-GAL4}attP2 | Bloomington *Drosophila* Stock Center | RRID:BDSC_48690 | *Jenett et al., 2012*; used in inc[2] rescue screen |
| Genetic reagent (*D. melanogaster*) | P{GMR15E09-GAL4}attP2 | Bloomington *Drosophila* Stock Center | RRID:BDSC_48696 | *Jenett et al., 2012*; used in inc[2] rescue screen |
| Genetic reagent (*D. melanogaster*) | P{GMR16E03-GAL4}attP2 | Bloomington *Drosophila* Stock Center | RRID:BDSC_48727 | *Jenett et al., 2012*; used in inc[2] rescue screen |
| Genetic reagent (*D. melanogaster*) | P{GMR17B12-GAL4}attP2 | Bloomington *Drosophila* Stock Center | RRID:BDSC_48759 | *Jenett et al., 2012*; used in inc[2] rescue screen |
| Genetic reagent (*D. melanogaster*) | P{GMR17D02-GAL4}attP2 | Bloomington *Drosophila* Stock Center | RRID:BDSC_48764 | *Jenett et al., 2012*; used in inc[2] rescue screen |
| Genetic reagent (*D. melanogaster*) | P{GMR17G05-GAL4}attP2 | Bloomington *Drosophila* Stock Center | RRID:BDSC_48782 | *Jenett et al., 2012*; used in inc[2] rescue screen |
| Genetic reagent (*D. melanogaster*) | P{GMR18D04-GAL4}attP2 | Bloomington *Drosophila* Stock Center | RRID:BDSC_48811 | *Jenett et al., 2012*; used in inc[2] rescue screen |
| Genetic reagent (*D. melanogaster*) | P{GMR18D07-GAL4}attP2 | Bloomington *Drosophila* Stock Center | RRID:BDSC_48813 | *Jenett et al., 2012*; used in inc[2] rescue screen |
| Genetic reagent (*D. melanogaster*) | P{GMR18F04-GAL4}attP2 | Bloomington *Drosophila* Stock Center | RRID:BDSC_48820 | *Jenett et al., 2012*; used in inc[2] rescue screen |
| Genetic reagent (*D. melanogaster*) | P{GMR18G06-GAL4}attP2 | Bloomington *Drosophila* Stock Center | RRID:BDSC_48826 | *Jenett et al., 2012*; used in inc[2] rescue screen |
| Genetic reagent (*D. melanogaster*) | P{GMR19F05-GAL4}attP2 | Bloomington *Drosophila* Stock Center | RRID:BDSC_48855 | *Jenett et al., 2012*; used in inc[2] rescue screen |

*Continued*

| Reagent type (species) or resource | Designation | Source or reference | Identifiers | Additional information |
|---|---|---|---|---|
| Genetic reagent (*D. melanogaster*) | P{GMR20F04-GAL4}attP2 | Bloomington *Drosophila* Stock Center | RRID:BDSC_48904 | *Jenett et al., 2012*; used in inc[2] rescue screen |
| Genetic reagent (*D. melanogaster*) | P{GMR21C09-GAL4}attP2 | Bloomington *Drosophila* Stock Center | RRID:BDSC_48936 | *Jenett et al., 2012*; used in inc[2] rescue screen |
| Genetic reagent (*D. melanogaster*) | P{GMR21D02-GAL4}attP2 | Bloomington *Drosophila* Stock Center | RRID:BDSC_48939 | *Jenett et al., 2012*; used in inc[2] rescue screen |
| Genetic reagent (*D. melanogaster*) | P{GMR21D06-GAL4}attP2 | Bloomington *Drosophila* Stock Center | RRID:BDSC_48942 | *Jenett et al., 2012*; used in inc[2] rescue screen |
| Genetic reagent (*D. melanogaster*) | P{GMR22C12-GAL4}attP2 | Bloomington *Drosophila* Stock Center | RRID:BDSC_48978 | *Jenett et al., 2012*; used in inc[2] rescue screen |
| Genetic reagent (*D. melanogaster*) | P{GMR22E06-GAL4}attP2 | Bloomington *Drosophila* Stock Center | RRID:BDSC_48986 | *Jenett et al., 2012*; used in inc[2] rescue screen |
| Genetic reagent (*D. melanogaster*) | P{GMR22H10-GAL4}attP2 | Bloomington *Drosophila* Stock Center | RRID:BDSC_49005 | *Jenett et al., 2012*; used in inc[2] rescue screen |
| Genetic reagent (*D. melanogaster*) | P{GMR23B04-GAL4}attP2 | Bloomington *Drosophila* Stock Center | RRID:BDSC_49016 | *Jenett et al., 2012*; used in inc[2] rescue screen |
| Genetic reagent (*D. melanogaster*) | P{GMR23C06-GAL4}attP2 | Bloomington *Drosophila* Stock Center | RRID:BDSC_49023 | *Jenett et al., 2012*; used in inc[2] rescue screen |
| Genetic reagent (*D. melanogaster*) | P{GMR23E10-GAL4}attP2 | Bloomington *Drosophila* Stock Center | RRID:BDSC_49032 | *Jenett et al., 2012*; used in inc[2] rescue screen |
| Genetic reagent (*D. melanogaster*) | P{GMR23F05-GAL4}attP2 | Bloomington *Drosophila* Stock Center | RRID:BDSC_49035 | *Jenett et al., 2012*; used in inc[2] rescue screen |
| Genetic reagent (*D. melanogaster*) | P{GMR24A08-GAL4}attP2 | Bloomington *Drosophila* Stock Center | RRID:BDSC_49058 | *Jenett et al., 2012*; used in inc[2] rescue screen |
| Genetic reagent (*D. melanogaster*) | P{GMR24B11-GAL4}attP2 | Bloomington *Drosophila* Stock Center | RRID:BDSC_49070 | *Jenett et al., 2012*; used in inc[2] rescue screen |
| Genetic reagent (*D. melanogaster*) | P{GMR24C06-GAL4}attP2 | Bloomington *Drosophila* Stock Center | RRID:BDSC_49073 | *Jenett et al., 2012*; used in inc[2] rescue screen |
| Genetic reagent (*D. melanogaster*) | P{GMR24C07-GAL4}attP2 | Bloomington *Drosophila* Stock Center | RRID:BDSC_49074 | *Jenett et al., 2012*; used in inc[2] rescue screen |
| Genetic reagent (*D. melanogaster*) | P{GMR24C10-GAL4}attP2 | Bloomington *Drosophila* Stock Center | RRID:BDSC_49075 | *Jenett et al., 2012*; used in inc[2] rescue screen |
| Genetic reagent (*D. melanogaster*) | P{GMR24E05-GAL4}attP2 | Bloomington *Drosophila* Stock Center | RRID:BDSC_49081 | *Jenett et al., 2012*; used in inc[2] rescue screen |

*Continued*

| Reagent type (species) or resource | Designation | Source or reference | Identifiers | Additional information |
|---|---|---|---|---|
| Genetic reagent (*D. melanogaster*) | P{GMR24F03-GAL4}attP2 | Bloomington *Drosophila* Stock Center | RRID:BDSC_49086 | *Jenett et al., 2012*; used in inc[2] rescue screen |
| Genetic reagent (*D. melanogaster*) | P{GMR24H03-GAL4}attP2 | Bloomington *Drosophila* Stock Center | RRID:BDSC_49098 | *Jenett et al., 2012*; used in inc[2] rescue screen |
| Genetic reagent (*D. melanogaster*) | P{GMR25A01-GAL4}attP2 | Bloomington *Drosophila* Stock Center | RRID:BDSC_49102 | *Jenett et al., 2012*; used in inc[2] rescue screen |
| Genetic reagent (*D. melanogaster*) | P{GMR25A06-GAL4}attP2 | Bloomington *Drosophila* Stock Center | RRID:BDSC_49105 | *Jenett et al., 2012*; used in inc[2] rescue screen |
| Genetic reagent (*D. melanogaster*) | P{GMR25C01-GAL4}attP2 | Bloomington *Drosophila* Stock Center | RRID:BDSC_49115 | *Jenett et al., 2012*; used in inc[2] rescue screen |
| Genetic reagent (*D. melanogaster*) | P{GMR25C03-GAL4}attP2 | Bloomington *Drosophila* Stock Center | RRID:BDSC_49117 | *Jenett et al., 2012*; used in inc[2] rescue screen |
| Genetic reagent (*D. melanogaster*) | P{GMR25E04-GAL4}attP2 | Bloomington *Drosophila* Stock Center | RRID:BDSC_49125 | *Jenett et al., 2012*; used in inc[2] rescue screen |
| Genetic reagent (*D. melanogaster*) | P{GMR25H06-GAL4}attP2 | Bloomington *Drosophila* Stock Center | RRID:BDSC_49144 | *Jenett et al., 2012*; used in inc[2] rescue screen |
| Genetic reagent (*D. melanogaster*) | P{GMR26B04-GAL4}attP2 | Bloomington *Drosophila* Stock Center | RRID:BDSC_49158 | *Jenett et al., 2012*; used in inc[2] rescue screen |
| Genetic reagent (*D. melanogaster*) | P{GMR26B11-GAL4}attP2 | Bloomington *Drosophila* Stock Center | RRID:BDSC_49164 | *Jenett et al., 2012*; used in inc[2] rescue screen |
| Genetic reagent (*D. melanogaster*) | P{GMR26B12-GAL4}attP2 | Bloomington *Drosophila* Stock Center | RRID:BDSC_49165 | *Jenett et al., 2012*; used in inc[2] rescue screen |
| Genetic reagent (*D. melanogaster*) | P{GMR26C11-GAL4}attP2 | Bloomington *Drosophila* Stock Center | RRID:BDSC_49171 | *Jenett et al., 2012*; used in inc[2] rescue screen |
| Genetic reagent (*D. melanogaster*) | P{GMR26E02-GAL4}attP2 | Bloomington *Drosophila* Stock Center | RRID:BDSC_49179 | *Jenett et al., 2012*; used in inc[2] rescue screen |
| Genetic reagent (*D. melanogaster*) | P{GMR26E07-GAL4}attP2 | Bloomington *Drosophila* Stock Center | RRID:BDSC_49182 | *Jenett et al., 2012*; used in inc[2] rescue screen |
| Genetic reagent (*D. melanogaster*) | P{GMR26F09-GAL4}attP2 | Bloomington *Drosophila* Stock Center | RRID:BDSC_49194 | *Jenett et al., 2012*; used in inc[2] rescue screen |
| Genetic reagent (*D. melanogaster*) | P{GMR27A02-GAL4}attP2 | Bloomington *Drosophila* Stock Center | RRID:BDSC_49207 | *Jenett et al., 2012*; used in inc[2] rescue screen |
| Genetic reagent (*D. melanogaster*) | P{GMR10E06-GAL4}attP2 | Bloomington *Drosophila* Stock Center | RRID:BDSC_49236 | *Jenett et al., 2012*; used in inc[2] rescue screen |

*Continued*

| Reagent type (species) or resource | Designation | Source or reference | Identifiers | Additional information |
|---|---|---|---|---|
| Genetic reagent (*D. melanogaster*) | P{GMR14B11-GAL4}attP2 | Bloomington *Drosophila* Stock Center | RRID:BDSC_49255 | *Jenett et al., 2012*; used in inc[2] rescue screen |
| Genetic reagent (*D. melanogaster*) | P{GMR15B03-GAL4}attP2 | Bloomington *Drosophila* Stock Center | RRID:BDSC_49261 | *Jenett et al., 2012*; used in inc[2] rescue screen |
| Genetic reagent (*D. melanogaster*) | P{GMR18G02-GAL4}attP2 | Bloomington *Drosophila* Stock Center | RRID:BDSC_49278 | *Jenett et al., 2012*; used in inc[2] rescue screen |
| Genetic reagent (*D. melanogaster*) | P{GMR32D08-GAL4}attP2 | Bloomington *Drosophila* Stock Center | RRID:BDSC_49357 | *Jenett et al., 2012*; used in inc[2] rescue screen |
| Genetic reagent (*D. melanogaster*) | P{GMR35F09-GAL4}attP2 | Bloomington *Drosophila* Stock Center | RRID:BDSC_49371 | *Jenett et al., 2012*; used in inc[2] rescue screen |
| Genetic reagent (*D. melanogaster*) | P{GMR60F05-GAL4}attP2 | Bloomington *Drosophila* Stock Center | RRID:BDSC_49405 | *Jenett et al., 2012*; used in inc[2] rescue screen |
| Genetic reagent (*D. melanogaster*) | P{GMR28E01-GAL4}attP2 | Bloomington *Drosophila* Stock Center | RRID:BDSC_49457 | *Jenett et al., 2012*; used in inc[2] rescue screen |
| Genetic reagent (*D. melanogaster*) | P{GMR29A12-GAL4}attP2 | Bloomington *Drosophila* Stock Center | RRID:BDSC_49478 | *Jenett et al., 2012*; used in inc[2] rescue screen |
| Genetic reagent (*D. melanogaster*) | P{GMR30B10-GAL4}attP2 | Bloomington *Drosophila* Stock Center | RRID:BDSC_49522 | *Jenett et al., 2012*; used in inc[2] rescue screen |
| Genetic reagent (*D. melanogaster*) | P{GMR43D09-GAL4}attP2 | Bloomington *Drosophila* Stock Center | RRID:BDSC_49553 | *Jenett et al., 2012*; used in inc[2] rescue screen |
| Genetic reagent (*D. melanogaster*) | P{GMR47E07-GAL4}attP2 | Bloomington *Drosophila* Stock Center | RRID:BDSC_49568 | *Jenett et al., 2012*; used in inc[2] rescue screen |
| Genetic reagent (*D. melanogaster*) | P{GMR48D07-GAL4}attP2 | Bloomington *Drosophila* Stock Center | RRID:BDSC_49572 | *Jenett et al., 2012*; used in inc[2] rescue screen |
| Genetic reagent (*D. melanogaster*) | P{GMR52F11-GAL4}attP2 | Bloomington *Drosophila* Stock Center | RRID:BDSC_49579 | *Jenett et al., 2012*; used in inc[2] rescue screen |
| Genetic reagent (*D. melanogaster*) | P{GMR59A05-GAL4}attP2 | Bloomington *Drosophila* Stock Center | RRID:BDSC_49593 | *Jenett et al., 2012*; used in inc[2] rescue screen |
| Genetic reagent (*D. melanogaster*) | P{GMR65H10-GAL4}attP2 | Bloomington *Drosophila* Stock Center | RRID:BDSC_49614 | *Jenett et al., 2012*; used in inc[2] rescue screen |
| Genetic reagent (*D. melanogaster*) | P{GMR66A03-GAL4}attP2 | Bloomington *Drosophila* Stock Center | RRID:BDSC_49615 | *Jenett et al., 2012*; used in inc[2] rescue screen |
| Genetic reagent (*D. melanogaster*) | P{GMR30G03-GAL4}attP2 | Bloomington *Drosophila* Stock Center | RRID:BDSC_49646 | *Jenett et al., 2012*; used in inc[2] rescue screen |

*Continued on next page*

*Continued*

| Reagent type (species) or resource | Designation | Source or reference | Identifiers | Additional information |
|---|---|---|---|---|
| Genetic reagent (*D. melanogaster*) | P{GMR31F06-GAL4}attP2 | Bloomington *Drosophila* Stock Center | RRID:BDSC_49684 | *Jenett et al., 2012*; used in inc[2] rescue screen |
| Genetic reagent (*D. melanogaster*) | P{GMR31G04-GAL4}attP2 | Bloomington *Drosophila* Stock Center | RRID:BDSC_49686 | *Jenett et al., 2012*; used in inc[2] rescue screen |
| Genetic reagent (*D. melanogaster*) | P{GMR31H05-GAL4}attP2 | Bloomington *Drosophila* Stock Center | RRID:BDSC_49692 | *Jenett et al., 2012*; used in inc[2] rescue screen |
| Genetic reagent (*D. melanogaster*) | P{GMR32E04-GAL4}attP2 | Bloomington *Drosophila* Stock Center | RRID:BDSC_49717 | *Jenett et al., 2012*; used in inc[2] rescue screen |
| Genetic reagent (*D. melanogaster*) | P{GMR33H07-GAL4}attP2 | Bloomington *Drosophila* Stock Center | RRID:BDSC_49760 | *Jenett et al., 2012*; used in inc[2] rescue screen |
| Genetic reagent (*D. melanogaster*) | P{GMR34B11-GAL4}attP2 | Bloomington *Drosophila* Stock Center | RRID:BDSC_49774 | *Jenett et al., 2012*; used in inc[2] rescue screen |
| Genetic reagent (*D. melanogaster*) | P{GMR34C08-GAL4}attP2 | Bloomington *Drosophila* Stock Center | RRID:BDSC_49780 | *Jenett et al., 2012*; used in inc[2] rescue screen |
| Genetic reagent (*D. melanogaster*) | P{GMR35B08-GAL4}attP2 | Bloomington *Drosophila* Stock Center | RRID:BDSC_49818 | *Jenett et al., 2012*; used in inc[2] rescue screen |
| Genetic reagent (*D. melanogaster*) | P{GMR10G02-GAL4}attP2 | Bloomington *Drosophila* Stock Center | RRID:BDSC_49825 | *Jenett et al., 2012*; used in inc[2] rescue screen |
| Genetic reagent (*D. melanogaster*) | P{GMR11E05-GAL4}attP2 | Bloomington *Drosophila* Stock Center | RRID:BDSC_49827 | *Jenett et al., 2012*; used in inc[2] rescue screen |
| Genetic reagent (*D. melanogaster*) | P{GMR19C10-GAL4}attP2 | Bloomington *Drosophila* Stock Center | RRID:BDSC_49831 | *Jenett et al., 2012*; used in inc[2] rescue screen |
| Genetic reagent (*D. melanogaster*) | P{GMR19E12-GAL4}attP2 | Bloomington *Drosophila* Stock Center | RRID:BDSC_49835 | *Jenett et al., 2012*; used in inc[2] rescue screen |
| Genetic reagent (*D. melanogaster*) | P{GMR20D07-GAL4}attP2 | Bloomington *Drosophila* Stock Center | RRID:BDSC_49848 | *Jenett et al., 2012*; used in inc[2] rescue screen |
| Genetic reagent (*D. melanogaster*) | P{GMR20E08-GAL4}attP2 | Bloomington *Drosophila* Stock Center | RRID:BDSC_49851 | *Jenett et al., 2012*; used in inc[2] rescue screen |
| Genetic reagent (*D. melanogaster*) | P{GMR21H06-GAL4}attP2 | Bloomington *Drosophila* Stock Center | RRID:BDSC_49866 | *Jenett et al., 2012*; used in inc[2] rescue screen |
| Genetic reagent (*D. melanogaster*) | P{GMR22F03-GAL4}attP2 | Bloomington *Drosophila* Stock Center | RRID:BDSC_49875 | *Jenett et al., 2012*; used in inc[2] rescue screen |
| Genetic reagent (*D. melanogaster*) | P{GMR35D07-GAL4}attP2 | Bloomington *Drosophila* Stock Center | RRID:BDSC_49908 | *Jenett et al., 2012*; used in inc[2] rescue screen |

*Continued*

| Reagent type (species) or resource | Designation | Source or reference | Identifiers | Additional information |
|---|---|---|---|---|
| Genetic reagent (*D. melanogaster*) | P{GMR37E08-GAL4}attP2 | Bloomington *Drosophila* Stock Center | RRID:BDSC_49958 | *Jenett et al., 2012*; used in inc[2] rescue screen |
| Genetic reagent (*D. melanogaster*) | P{GMR37F05-GAL4}attP2 | Bloomington *Drosophila* Stock Center | RRID:BDSC_49961 | *Jenett et al., 2012*; used in inc[2] rescue screen |
| Genetic reagent (*D. melanogaster*) | P{GMR38A11-GAL4}attP2 | Bloomington *Drosophila* Stock Center | RRID:BDSC_49980 | *Jenett et al., 2012*; used in inc[2] rescue screen |
| Genetic reagent (*D. melanogaster*) | P{GMR38B06-GAL4}attP2 | Bloomington *Drosophila* Stock Center | RRID:BDSC_49986 | *Jenett et al., 2012*; used in inc[2] rescue screen |
| Genetic reagent (*D. melanogaster*) | P{GMR38E08-GAL4}attP2 | Bloomington *Drosophila* Stock Center | RRID:BDSC_50008 | *Jenett et al., 2012*; used in inc[2] rescue screen |
| Genetic reagent (*D. melanogaster*) | P{GMR39C07-GAL4}attP2 | Bloomington *Drosophila* Stock Center | RRID:BDSC_50039 | *Jenett et al., 2012*; used in inc[2] rescue screen |
| Genetic reagent (*D. melanogaster*) | P{GMR39E10-GAL4}attP2 | Bloomington *Drosophila* Stock Center | RRID:BDSC_50053 | *Jenett et al., 2012*; used in inc[2] rescue screen |
| Genetic reagent (*D. melanogaster*) | P{GMR39G09-GAL4}attP2 | Bloomington *Drosophila* Stock Center | RRID:BDSC_50064 | *Jenett et al., 2012*; used in inc[2] rescue screen |
| Genetic reagent (*D. melanogaster*) | P{GMR40C07-GAL4}attP2 | Bloomington *Drosophila* Stock Center | RRID:BDSC_50080 | *Jenett et al., 2012*; used in inc[2] rescue screen |
| Genetic reagent (*D. melanogaster*) | P{GMR42D11-GAL4}attP2 | Bloomington *Drosophila* Stock Center | RRID:BDSC_50156 | *Jenett et al., 2012*; used in inc[2] rescue screen |
| Genetic reagent (*D. melanogaster*) | P{GMR44B03-GAL4}attP2 | Bloomington *Drosophila* Stock Center | RRID:BDSC_50200 | *Jenett et al., 2012*; used in inc[2] rescue screen |
| Genetic reagent (*D. melanogaster*) | P{GMR44B10-GAL4}attP2 | Bloomington *Drosophila* Stock Center | RRID:BDSC_50202 | *Jenett et al., 2012*; used in inc[2] rescue screen |
| Genetic reagent (*D. melanogaster*) | P{GMR44D02-GAL4}attP2 | Bloomington *Drosophila* Stock Center | RRID:BDSC_50205 | *Jenett et al., 2012*; used in inc[2] rescue screen |
| Genetic reagent (*D. melanogaster*) | P{GMR45D05-GAL4}attP2 | Bloomington *Drosophila* Stock Center | RRID:BDSC_50227 | *Jenett et al., 2012*; used in inc[2] rescue screen |
| Genetic reagent (*D. melanogaster*) | P{GMR45G01-GAL4}attP2 | Bloomington *Drosophila* Stock Center | RRID:BDSC_50241 | *Jenett et al., 2012*; used in inc[2] rescue screen |
| Genetic reagent (*D. melanogaster*) | P{GMR45G05-GAL4}attP2 | Bloomington *Drosophila* Stock Center | RRID:BDSC_50243 | *Jenett et al., 2012*; used in inc[2] rescue screen |
| Genetic reagent (*D. melanogaster*) | P{GMR45H11-GAL4}attP2 | Bloomington *Drosophila* Stock Center | RRID:BDSC_50248 | *Jenett et al., 2012*; used in inc[2] rescue screen |

*Continued*

| Reagent type (species) or resource | Designation | Source or reference | Identifiers | Additional information |
|---|---|---|---|---|
| Genetic reagent (*D. melanogaster*) | P{GMR46B05-GAL4}attP2 | Bloomington *Drosophila* Stock Center | RRID:BDSC_50253 | *Jenett et al., 2012*; used in inc[2] rescue screen |
| Genetic reagent (*D. melanogaster*) | P{GMR47D07-GAL4}attP2 | Bloomington *Drosophila* Stock Center | RRID:BDSC_50304 | *Jenett et al., 2012*; used in inc[2] rescue screen |
| Genetic reagent (*D. melanogaster*) | P{GMR47F04-GAL4}attP2 | Bloomington *Drosophila* Stock Center | RRID:BDSC_50319 | *Jenett et al., 2012*; used in inc[2] rescue screen |
| Genetic reagent (*D. melanogaster*) | P{GMR47G08-GAL4}attP2 | Bloomington *Drosophila* Stock Center | RRID:BDSC_50328 | *Jenett et al., 2012*; used in inc[2] rescue screen |
| Genetic reagent (*D. melanogaster*) | P{GMR47H01-GAL4}attP2 | Bloomington *Drosophila* Stock Center | RRID:BDSC_50330 | *Jenett et al., 2012*; used in inc[2] rescue screen |
| Genetic reagent (*D. melanogaster*) | P{GMR48A03-GAL4}attP2 | Bloomington *Drosophila* Stock Center | RRID:BDSC_50339 | *Jenett et al., 2012*; used in inc[2] rescue screen |
| Genetic reagent (*D. melanogaster*) | P{GMR48A08-GAL4}attP2 | Bloomington *Drosophila* Stock Center | RRID:BDSC_50341 | *Jenett et al., 2012*; used in inc[2] rescue screen |
| Genetic reagent (*D. melanogaster*) | P{GMR48B10-GAL4}attP2 | Bloomington *Drosophila* Stock Center | RRID:BDSC_50352 | *Jenett et al., 2012*; used in inc[2] rescue screen |
| Genetic reagent (*D. melanogaster*) | P{GMR48C06-GAL4}attP2 | Bloomington *Drosophila* Stock Center | RRID:BDSC_50357 | *Jenett et al., 2012*; used in inc[2] rescue screen |
| Genetic reagent (*D. melanogaster*) | P{GMR48E02-GAL4}attP2 | Bloomington *Drosophila* Stock Center | RRID:BDSC_50367 | *Jenett et al., 2012*; used in inc[2] rescue screen |
| Genetic reagent (*D. melanogaster*) | P{GMR48G01-GAL4}attP2 | Bloomington *Drosophila* Stock Center | RRID:BDSC_50381 | *Jenett et al., 2012*; used in inc[2] rescue screen |
| Genetic reagent (*D. melanogaster*) | P{GMR48G04-GAL4}attP2 | Bloomington *Drosophila* Stock Center | RRID:BDSC_50383 | *Jenett et al., 2012*; used in inc[2] rescue screen |
| Genetic reagent (*D. melanogaster*) | P{GMR48H04-GAL4}attP2 | Bloomington *Drosophila* Stock Center | RRID:BDSC_50392 | *Jenett et al., 2012*; used in inc[2] rescue screen |
| Genetic reagent (*D. melanogaster*) | P{GMR48H10-GAL4}attP2 | Bloomington *Drosophila* Stock Center | RRID:BDSC_50395 | *Jenett et al., 2012*; used in inc[2] rescue screen |
| Genetic reagent (*D. melanogaster*) | P{GMR48H11-GAL4}attP2 | Bloomington *Drosophila* Stock Center | RRID:BDSC_50396 | *Jenett et al., 2012*; used in inc[2] rescue screen |
| Genetic reagent (*D. melanogaster*) | P{GMR49A09-GAL4}attP2 | Bloomington *Drosophila* Stock Center | RRID:BDSC_50403 | *Jenett et al., 2012*; used in inc[2] rescue screen |
| Genetic reagent (*D. melanogaster*) | P{GMR49C03-GAL4}attP2 | Bloomington *Drosophila* Stock Center | RRID:BDSC_50414 | *Jenett et al., 2012*; used in inc[2] rescue screen |

## Fly food and culture

Fly food was prepared in batches containing the following ingredients: 1800 g cornmeal (Labscientific, FLY-8010-20), 1800 ml molasses (Labscientific, FLY-8008-16), 744 g yeast (Labscientific, FLY-8040-20F),

266 g agar (Mooragar, 41084), 56 g methyl paraben (Sigma, H3647), 560 ml alcohol (Fisher, A962P4), 190 ml propionic acid (Fisher, A258500), and 47 l of water. Unless indicated otherwise, crosses were performed with five females and three males in vials (28.5 mm diameter × 95 mm height) containing standard fly food supplemented with dry yeast (Fleischmann, B000LRFVHE). Crosses were cultured at 25°C in 12 hr light–dark (LD) cycles.

To prepare food for conditional induction of the Q-system, solid fly food was melted in a microwave oven and allowed to cool before addition of quinic acid or vehicle. Quinic acid solution was freshly prepared essentially as described (*Riabinina et al., 2015*). 10 g of quinic acid (Sigma, 138622) was dissolved in 30 ml of water and the pH was adjusted to 6.5 with 10 mM NaOH. A volume of quinic acid solution containing the equivalent of 0.66 g of quinic acid (~2.4 ml) was added for each 10 ml of melted fly food and mixed well; ~12.4 ml was distributed to each empty vial. Food was allowed to cool and subsequently stored at 4°C prior to use. Vehicle food was prepared similarly, substituting an equal volume of water.

## Conditional Q-system induction

Three sets of conditional induction experiments were performed. The first set contained vehicle treatment and constitutive, developmental-specific, and adult-specific induction regimens. The second set included vehicle, constitutive induction, and induction from the late third instar larval stage through adulthood. The third set included vehicle, constitutive induction, and a pulse of induction from the late third instar larval stage through pupal stages. Initiation, maintenance, or termination of induction at desired developmental stages was achieved by transferring larvae, pupae, and/or adults to food containing quinic acid or vehicle as described below. Within each set of experiments, $w^{1118}$ and $inc^1$ controls were exposed to vehicle and quinic acid induction regimens, and all animals underwent the same physical transfers in parallel. Sleep of $w^{1118}$ and $inc^1$ animals was not altered by exposure to vehicle or quinic acid, as described previously (*Li and Stavropoulos, 2016*), nor by physical transfer at larval, pupal, or adult stages. Vehicle-treated $w^{1118}$ and $inc^1$ animals, pooled across all three sets of experiments, are shown in *Figure 2B*. Two to three independent biological replications were performed for all induction experiments.

In the first set of experiments, developmental-specific induction was achieved by setting crosses on food containing quinic acid, allowing animals to develop and pupate in the same vials, and transferring adult males within 2–3 hr of eclosion to fresh vials with vehicle-containing food to terminate Q-system induction. Adult animals were maintained in these vials for 3–4 days, anesthetized with $CO_2$, and transferred to DAM tubes with vehicle-containing food for measurement of sleep. For adult-specific induction, crosses were set on vehicle food and animals developed in the same vials. Adult males eclosing from these cultures were transferred within 2–3 hr of eclosion to fresh vials with food containing quinic acid, maintained on this food for 3–4 days, and transferred to DAM tubes containing food with quinic acid for measurement of sleep. For constitutive induction and vehicle treatment, food containing quinic acid or vehicle, respectively, was used throughout, along with the same transfer procedure.

In the second set of experiments, induction from the late third instar larval stage through adulthood was achieved as follows: crosses were set on vehicle-containing food and wandering third instar larvae from these cultures were gently collected with blunt forceps and examined under brief phosphate-buffered saline (PBS) immersion to select males by visual identification of gonads as described (*Kerkis, 1931*). Larvae were transferred to recipient vials containing isogenic $w^{1118}$ larvae and pre-churned quinic acid food; these recipient cultures were initiated in parallel with experimental crosses to allow food consistency to be maintained during Q-system induction. Adult animals bearing *mini-white*-marked transgenes were transferred within 2–3 hr of eclosion to fresh vials containing quinic acid food to maintain Q-system induction. Three- to four-day-old adults were subsequently transferred to DAM tubes with food containing quinic acid for measurement of sleep. Constitutive induction and vehicle treatment were performed similarly, using appropriate food and the same transfer procedure.

In the third set of experiments, a pulse of Q-system induction specific to late third instar larval and pupal stages was achieved as follows: crosses were set on vehicle food and male wandering third instar larval progeny were selected and transferred to $w^{1118}$ recipient vials containing pre-churned quinic acid food as described above. To prevent adult exposure to quinic acid, pupae bearing *mini-white*-marked transgenes were identified at approximately the P13–P14 stage by pigmented eyes and

black wings (*Ashburner et al., 2005*; *Bainbridge and Bownes, 1981*) and gently dislodged from vial walls with a paintbrush and transferred to the walls of fresh vials containing vehicle food. Three- to four-day-old adults eclosing from these vials were transferred to DAM tubes containing vehicle food for measurements of sleep. Constitutive induction and vehicle treatment were performed similarly, using appropriate food and the same transfer procedure.

### $inc^2$ rescue screen

$inc^2$ virgins were crossed to male flies carrying Gal4 transgenes and a minimum of five male progeny were screened for each genotype. A total of 277 Gal4 lines were screened, including 266 randomly selected drivers and 11 drivers previously characterized for expression in sleep-regulatory circuits. To select random lines, 4088 lines from the FlyLight collection available from the Bloomington *Drosophila* Stock Center were assigned sample numbers. Using the *randperm* command in Matlab, 300 lines were randomly selected. Expression patterns for these lines in the Janelia Flylight database were examined; 84 lines were excluded due to very low levels of expression, very broad expression patterns unlikely to be useful for functional mapping, or because expression data were unavailable. Expression patterns for the remaining 216 lines ranged from broad to sparse. This procedure for random selection was applied iteratively to yield 266 lines. Top-ranking hits from the initial screen were rescreened in independent crosses. Rescreening of c253-Gal4 and c309-Gal4 was performed after backcrossing each line six generations to an isogenic $w^{1118}$ stock (BDSC #5905) (*Ryder et al., 2004*).

### MB ablation

MB ablation was performed essentially as described previously (*de Belle and Heisenberg, 1994*). Egg collection was performed on grape juice agar plates containing a spot of rehydrated dry yeast. $w^{1118}$ larvae at the first instar stage were transferred to a well of a 24-well plate bearing a spot of rehydrated dry yeast paste, containing water vehicle or 50 mg/ml hydroxyurea (Sigma, H8627). After 4–5 hr, larvae were collected and washed briefly with distilled water on a Nitex mesh filter (Genesee Scientific, 57–102) to remove yeast and subsequently transferred to vials containing standard food. Vials were cultured at 25°C in LD cycles and adult animals eclosing from these cultures were assayed for sleep as described below. MB ablation was verified in adult brains in a separate cohort of animals by staining with anti-FasII primary antibody (1:50, DSHB) and Alexa 488-conjugated donkey anti-mouse secondary as described below. Vehicle-treated animals exhibited MB lobes demarcated with FasII signal (100%, $n = 9$), while hydroxyurea-treated animals exhibited complete MB ablation as indicated by the lack of residual FasII staining (100%, $n = 12$); FasII signal within the EB was observed in all brains, providing a control for staining of the MB.

### Immunohistochemistry

All fixing, washing, and incubation steps for immunohistochemistry were performed on a nutator. To assess conditional induction of *inc*-HA using the Q-system, larval, pupal, and adult brains were dissected from $inc^1$; *UAS-inc-HA/tub-QS*; *nsyb-Gal4QF/+* males. Wandering third instar male larvae were selected by visual identification of gonads as described above. Larval brains were dissected in ice-cold PBS, fixed with 4% paraformaldehyde in PBS for 30 min at room temperature, and washed 3× 15 min in PBS containing 0.2% Triton X-100 (PBST). Male pupae at stage P13–P14 were identified by the staging criteria described above and the presence of sex combs. Pupal brains were dissected in ice-cold PBST, fixed with 4% paraformaldehyde in PBST for 30 min at room temperature, and washed 3× 15 min in PBST. To prepare adult brains, 2- or 4-day-old whole male adults were fixed with 4% paraformaldehyde in PBST for 3 hr at 4°C and washed 3× 15 min in PBST at room temperature prior to brain dissection in PBST. After dissection, all brains were blocked with 5% normal donkey serum (NDS) (Lampire Biological, 7332500) in PBST at room temperature for 30–60 min. Samples were incubated overnight at 4°C in rat anti-HA (1:100; Sigma, 11867431001) and mouse anti-Brp (1:20, DSHB, nc82) antibodies prepared in 5% NDS in PBST. Brains were subsequently washed 3× 15 min in PBST at room temperature, incubated overnight at 4°C in Alexa 488 donkey anti-rat (1:1000; Life Technologies, A21208) and Alexa 647 donkey anti-mouse (1:1000, Life Technologies A31571) antibodies prepared in 5% NDS in PBST, washed 3× 15 min at room temperature in PBST, and mounted on microscope slides (Fisher, 1255015) in Vectashield (Vector Labs, H-1000).

For all other immunohistochemistry, adult brains of 4-day-old males were dissected in PBST, fixed with 4% paraformaldehyde in PBST for 30 min at room temperature, and washed 3× 20 min in PBST at room temperature. Brains of male wandering third instar larvae and stage P13–P14 pupae were dissected, fixed, and stained as described above for Q-system experiments. Primary antibodies were mouse anti-FLAG (1:100; Sigma, F1804), rabbit anti-GFP (1:2000; Fisher, A11122), mouse anti-GFP (1:1000; DSHB, GFP-G1), rabbit anti-dsRed (1:1000; Takara, 632496), and mouse anti-Brp (1:50, DSHB, nc82). Secondary antibodies were Alexa 488 donkey anti-rabbit (1:1000; Life Technologies, A21206), Alexa 488 donkey anti-mouse (1:1000; Life Technologies, A21202), and Alexa 568 donkey anti-rabbit (1:1000; Life Technologies, A10042).

## Imaging and quantitation of neuron number and cluster number

All imaging was performed on a Zeiss LSM800 confocal microscope, using a 10X air objective to capture z-stacks at 512 × 512 pixel resolution with 1 μM z-slices, unless indicated otherwise. All imaging settings were identical for each experiment comprising control and experimental brains stained in parallel.

To quantify MB neuron numbers, wild-type and $inc^1$ brains expressing *UAS-MyrGFP-2A-RedStinger* under the control of split-Gal4 drivers were imaged as described above. For each neuron subtype, wild-type and $inc^1$ brains were assigned sample numbers and a subset, randomly selected using the *randperm* command in Matlab, was imaged at higher resolution with a 63X oil objective. Both hemispheres of brains were imaged, capturing dsRed and myr-GFP channels separately. Only a single hemisphere could be imaged for two wild-type brains, one each in the $\gamma_d$ and $\alpha'/\beta'$ groups, due to sample compression by the objective. z-stacks encompassing nuclei were captured at 512 × 512 resolution for $\gamma_d$ neurons and at 1024 × 1024 resolution for $\alpha'/\beta'$ and $\alpha/\beta_c$ neurons; 2 μM z-slices were used to ensure that all nuclei (diameter ~3 μM) were segmented in at least one optical section.

High resolution z-stacks were assigned a random letter code and neurons were counted in a single-blind manner by two independent experimenters. Nuclei of $\gamma_d$ and $\alpha'/\beta'$ neurons exhibited minimal overlap along the z-axis, allowing nuclei to be counted in maximum intensity z-projections using the Cell Counter plug-in in ImageJ; visual inspection of z-stacks in parallel allowed overlapping nuclei to be differentiated. Dense distribution of $\alpha/\beta_c$ neurons prohibited accurate counting in single maximum intensity z-projections; maximum intensity z-projections were generated for every 10 z-slices, yielding three to four maximum intensity z-projections representing 20 μM each. To improve visualization of densely clustered $\alpha/\beta_c$ nuclei, background was subtracted using a rolling ball/sliding paraboloid algorithm (radius set to the size of the largest nucleus: 50 pixels) and image intensity display range was adjusted (minimum: 5; maximum: 175). Processed maximum intensity z-projections representing 20 μM each were then merged into a single z-stack for manual counting using the Cell Counter plug-in in ImageJ; to avoid double-counting of nuclei segmented in adjacent z-projections, the original unprocessed z-stack was examined in parallel. The variation in MB neuron counts between experimenters, calculated as the absolute difference between the two counts divided by their mean, was (mean ± SEM) 2.0% ± 0.2% for $\gamma_d$; 1.5% ± 0.3% for $\alpha'/\beta'$; and 2.6% ± 0.3% for $\alpha/\beta_c$. Where neuron counts were different for a given hemisphere, the average was plotted. Numbers of $\gamma_d$ neurons in wild-type animals were intermediate between those reported in prior studies (*Aso et al., 2014a*; *Shih et al., 2019*), while numbers of $\alpha'/\beta'$ and $\alpha/\beta_c$ neurons were lower, likely reflecting conservative assignment of nuclei in our study and the use of different antibodies and reporters (*Aso et al., 2014a*; *Shih et al., 2019*). $\alpha'/\beta'$ counts obtained using the MB370B driver were similar to previously reported numbers of $\alpha'/\beta'_m$ neurons; because MB370B labels $\alpha'/\beta'_m$ neurons strongly and $\alpha'/\beta'_{ap}$ neurons weakly, the lower absolute numbers of $\alpha'/\beta'$ neurons in our studies may reflect detection sensitivity and correspond chiefly to $\alpha'/\beta'_m$ neurons.

To count the number of $\alpha/\beta_c$ neuron clusters in wild-type and $inc^1$ brains, the same randomly selected samples used to quantify neuron numbers were assessed in a single-blind manner by two independent experimenters. Each z-stack was analyzed using a combination of visual inspection of z-sections and rotating the image stack in three dimensions using the 3D Viewer plug-in in ImageJ (threshold: 0; resampling factor: 2). A group of nuclei distributed continuously along all axes was classified as a cluster; a continuous gap at least one nuclear diameter in width across all axes was used to define cluster edges and discrete clusters. Cluster counts were identical for wild-type brains; total

cluster counts for $inc^1$ brains differed by 8.2% ± 0.2% (mean ± SEM) between experimenters. Where cluster counts were different for a given hemisphere, the average was plotted.

DH44 and dFB somata were counted in a single-blind manner by two independent experimenters as described for $\gamma_d$ and $\alpha'/\beta'$ neurons, using *DH44-Gal4* to drive *UAS-MyrGFP-2A-RedStinger* and *23E10-Gal4* to drive *5× UAS-IVS-Syt1::smGdP-HA*. Numbers of dFB and DH44 neurons were identical between two independent experimenters.

## Analysis of axonal projections and dendritic volume

To analyze axonal projections and dendritic volume, image stacks were captured using a 10X objective at 512 × 512 resolution with 1 μM *z*-slices. Axonal projection defects were assessed in maximum intensity z-projections. The number of horizontal and/or vertical lobes missing myr-GFP signal entirely was counted for each brain. To quantify dendritic volume, the Threshold command in ImageJ was applied to *z*-stacks to select dendrites based on DenMark immunofluorescence; high signal to noise allowed unambiguous demarcation of dendrites and clear separation from background. The same minimum and maximum threshold values were applied to all wild-type and $inc^1$ brains stained in parallel in an experiment and captured the entirety of dendritic signal for all samples. A single rectangular region of interest of minimal area encompassing dendritic signals from both brain hemispheres across all *z*-slices was drawn for each *z*-stack. Dendritic volume was quantified using the Voxel Counter plug-in in ImageJ.

## Sleep analysis

Three- to four-day-old male flies eclosing from LD-entrained cultures raised at 25°C were loaded in glass tubes (5 mm diameter × 65 mm length) containing standard food or appropriate food for Q-system experiments as described above. Animals were monitored for 5–7 days at 25°C in LD cycles using DAM2 monitors (Trikinetics). Locomotor activity data were collected in 1 min bins. Inactive periods of 5 min or longer were classified as sleep. The first 36–48 hr of data were discarded to allow acclimation of animals to tubes, and 3–5 integral days of data were analyzed beginning with ZT0. Dead animals were excluded from analysis by a combination of automated filtering and visual inspection of locomotor traces. Matlab code used to analyze sleep is available in *Source code 1*.

## Thermogenetic activation

Crosses were set on standard fly food as described above and cultured at 21.5°C. One- to four-day-old male flies eclosing from these cultures were assayed for 5 days in LD cycles. Animals were maintained at 21.5°C for the first 60–72 hr of the assay, including 36–48 hr of acclimation and the subsequent baseline day beginning at ZT0. Temperature was increased to 28.5°C for 24 hr to activate dTrpA1, followed by 24 hr of recovery at 21.5°C. The percent change in sleep was calculated for each animal by subtracting the amount of sleep on the baseline day from the amount of sleep on the activation day and dividing this difference by the amount of sleep on the baseline day. The percent change in sleep for individual animals was averaged for each genotype.

Gal4 drivers used to express TrpA1 were as follows: pan-MB, *MB004B split-Gal4*; MB, *c253-Gal4* and *c309-Gal4*; EB, *R69F08-Gal4*; DPM, *NP2721-Gal4*; Helicon, *R24B11-Gal4*; l-LNV, *c929-Gal4*; PI, PPM3, *c584-Gal4*.

## Statistics

One-way analysis of variance (ANOVA) and Tukey post hoc tests were used for comparisons between more than two groups of animals for total sleep, daytime sleep, nighttime sleep, and sleep bout number; for comparisons of these sleep parameters between two groups, unpaired two-sided Student's *t*-tests were used. Kruskal–Wallis tests and Dunn's post hoc tests were used for comparisons of sleep bout length between more than two groups of animals; for comparison between two groups, Mann–Whitney tests were used. One-way ANOVA and Dunnett's post hoc tests were used for

comparisons of percent change in sleep. Unpaired two-sided Welch's *t*-tests were used for pairwise comparisons of neuron number, cluster number, and dendrite volume.

## Acknowledgements

We thank C Desplan, N Ringstad, M Shirasu-Hiza, and members of the Stavropoulos lab for comments on the manuscript; Y Aso, B Ganetzky, L Griffith, W Joiner, W Li, K Nagel, C Potter, A Sehgal, J Simpson, G Suh, M Wu, M Young, the Bloomington *Drosophila* Stock Center, the Vienna *Drosophila* Resource Center, and the Janelia Flylight collection for fly stocks; and DSHB for antibodies. This work was supported by an International Student Research Fellowship from the Howard Hughes Medical Institute (HHMI) to QL and by grants from the National Institutes of Health (R01NS112844 and R21NS111304), the Mathers Foundation, Whitehall Foundation grant 2013-05-78, fellowships from the Alfred P Sloan and Leon Levy Foundations, a NARSAD Young Investigator Award from the Brain and Behavior Foundation, the J Christian Gillin, M.D. Research Award from the Sleep Research Society Foundation, and a Career Scientist Award from the Irma T Hirschl/Weill-Caulier Trust to NS.

## Additional information

### Funding

| Funder | Grant reference number | Author |
|---|---|---|
| Howard Hughes Medical Institute | International Student Research Fellowship | Qiuling Li |
| National Institute of Neurological Disorders and Stroke | R01NS112844 | Nicholas Stavropoulos |
| National Institute of Neurological Disorders and Stroke | R21NS111304 | Nicholas Stavropoulos |
| G. Harold and Leila Y. Mathers Foundation | | Nicholas Stavropoulos |
| Whitehall Foundation | 2013-05-78 | Nicholas Stavropoulos |
| Alfred P. Sloan Foundation | | Nicholas Stavropoulos |
| Leon Levy Foundation | | Nicholas Stavropoulos |
| Brain and Behavior Research Foundation | NARSAD Young Investigator | Nicholas Stavropoulos |
| Sleep Research Society Foundation | J. Christian Gillin, M.D. Research Grant | Nicholas Stavropoulos |
| Irma T. Hirschl Trust | Career Scientist Award | Nicholas Stavropoulos |

The funders had no role in study design, data collection, and interpretation, or the decision to submit the work for publication.

### Author contributions

Qiuling Li, Conceptualization, Investigation, Methodology, Writing – original draft, Writing – review and editing; Hyunsoo Jang, Kayla Y Lim, Alexie Lessing, Investigation, Methodology, Writing – review and editing; Nicholas Stavropoulos, Conceptualization, Funding acquisition, Project administration, Software, Supervision, Writing – original draft, Writing – review and editing

### Author ORCIDs

Qiuling Li http://orcid.org/0000-0002-6101-8779
Hyunsoo Jang http://orcid.org/0000-0001-9191-3697
Kayla Y Lim http://orcid.org/0000-0001-7168-5877
Alexie Lessing http://orcid.org/0000-0003-2044-7822
Nicholas Stavropoulos http://orcid.org/0000-0001-5915-2760

Decision letter and Author response
Decision letter https://doi.org/10.7554/eLife.65437.sa1
Author response https://doi.org/10.7554/eLife.65437.sa2

## Additional files

### Supplementary files
- Transparent reporting form
- Source data 1. Data for all figure panels.
- Source code 1. Matlab code for analysis of sleep.

### Data availability
Data for all figures and code used to analyze sleep are included in the supporting files.

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
