## [Editor Report]

This is an interesting study showing that the short sleep phenotype of *inc* mutants in *Drosophila* depends on the loss of the gene at a specific developmental time, and in a specific region, the mushroom bodies (MB). There are very few studies assessing the effects of sleep during development, in any animal species, and thus this paper is a very welcomed addition. The experiments are carefully done, and the conclusions are warranted.

---

## [Decision Letter]

**Decision letter after peer review:**

Thank you for submitting your article "*insomniac* links the development and function of a sleep regulatory circuit" for consideration by *eLife*. Your article has been reviewed by 3 peer reviewers, and the evaluation has been overseen by a Reviewing Editor and Ronald Calabrese as the Senior Editor. The following individual involved in review of your submission has agreed to reveal their identity: Chiara Cirelli (Reviewer #2).

Essential revisions:

1) The authors should briefly discuss *inc*'s previously described role in homeostatic plasticity, which may be relevant to this manuscript. Perhaps MBs require homeostatic plasticity to promote sleep but they cannot undergo it without *inc* In that case, multiplication and rewiring of MB neurons may not be cell-autonomous direct effects of reducing *inc*'s molecular function but instead may be compensatory responses (i.e. expansion and rewiring of sleep-promoting neurons) to make up for loss of sleep that begins to manifest in larvae. Hypofunctionality of MB neurons in this context might thus be expected to reduce sleep just like loss of MB neurons following ablation with hydroxyurea. Previous studies indicate that hydroxyurea ablation of the MBs phenocopies *inc* mutants' reduction in sleep. A comparison between these two short-sleeping phenotypes should be made and can be accomplished using ablation data already published.

2) The authors should add at least a few lines of commentary to the *banderuola* (*bnd*) gene. Bnd controls asymmetric cell division of sensory organ neural precursors. 5' *wake* alleles are derived from *bnd*, and these alleles reduce sleep, like LOF mutants of *inc* (Mauri et al., Curr Biol 2014; Liu et al., Neuron 2014; see also Zhang et al., PLoS Genet 2015). Those findings bolster the contention in the current study that the function of many sleep-regulating genes may be developmental in origin.

3) There were concerns related to how the data is presented, how some conclusions are made and why some critically important data is left out. Several suggesting for improvement in these regards are in the reviewers' "Recommendations for the authors." Some important points to address:

a. In Figures 1-5, sleep data is only represented as a sleep profile or total sleep. Sleep is a complex behavior and when we manipulate genes several parameters are affected. Regarding sleep parameters it would be great if authors can be consistent. For example, figure 4-supplement 1 has all important sleep parameters for the Mb sleep rescue experiment but the same parameters are not shown for other experiments (e.g. quinic acid). If there is a specific reason for presenting additional parameters for one set of experiments and not the others, then it should be made explicit. It appears that authors think these parameters are important because they show them for this one set of experiments. Also, these experiments are different from the published Neuron 72:964-976. doi:10.1016/j.neuron.2011.12.003 paper showing *inc* mutants and these manipulations could affect sleep parameters differently.

b. There is a screen of 283 Gal4 lines in Figure 4A. The lines are listed by Bloomington id but the genotype information is absent from material and methods (copy and paste genotype information from Bloomington). We are interested in information like 49E09-Gal4 instead of BL38692 (Bloomington id) in the Materials and methods section. The text says sleep regulatory circuits or randomly selected without any additional details. How many were random, how many were sleep regulatory and where, were they clean or mixed population?

c. Further, the figures define several transgenic lines as pan-MB gal4, DPM-gal4 etc. (Figure 5). Several gal4 lines could have a DPM neuron and many other neurons so it potentially confusing to call each a DPM Gal4. Please make it clear to the reader what Gal4 was used and where its expressed. This can be done simply in the figure legend and in a supplementary table with appropriate references to the literature so people can see for themselves where a certain line is expressed by going to the appropriate reference. This would also help people doing similar RNAi screens.

*Reviewer #1:*

The submitted manuscript explores an interesting area in neuroscience as to how genes control development of circuits that regulate behavior. What do these genes do in these circuits and at what stage do they become critical. Authors focus on 3 main things:

1) Where *inc* is expressed and when its required for circuit development that underlies *inc* dependent sleep regulation.

2) *inc* expression in MB and how that controls sleep.

3) How *inc* expression regulates development of MB-KCs and assemble into functional circuits.

These findings are important for the field and experiments are appropriate and figures are generally clear. My major concerns are related to how the data is presented, how some conclusions are made and why some critically important data is left out. Figure 1-5, sleep data is only represented as a sleep profile or total sleep. Sleep is a complex behavior and when we manipulate genes several parameters are affected. These sleep parameters should be shown for all genetic and other (quinic acid) manipulations.

The screening of 283 GAL4 drivers for *inc* expression and its affect on sleep is shown but that data is barely discussed. If the intention is not to discuss the data then its positioning in main figures should be reconsidered as it is not clear what that screen is all about.

The effect of *inc* expression on KC cell number is intriguing but there is no direct evidence that reduction in the percentage of KC population affects sleep. Although, *inc* is expressed in MBs and *inc* is involved in KC neurogenesis but how do these flies with defective KCs behave. Is it the number of KCs that regulate sleep or molecular role of *inc* in MB functioning/ physiological activity?

Since this paper focusses on the role of a single gene, so KCs with *inc* knockdown could be physiologically altered suggesting that while *inc* is involved in neurogenesis it is not necessarily the reason flies sleep differently. What this gene does to circuits and connectivity is a key focus of the paper but the experiments do directly address this. what is the significance of KC number to sleep and how are specific KCs related to the *inc* phenotypes? This needs to be addressed thoroughly.

Specific Comments

Introduction: Since gene expression is the big focus of this paper the authors need to elaborate on what specific genes have been identified and how they control sleep circuits. While, the focus is *Drosophila* sleep it might be worthwhile mentioning some genes and how they shape circuits in other animals. This is an important area in sleep research that has not received much focus but the central problem and specifically role of *inc* and *Cul3* gene need to be significantly elaborated.

Results section: Lines 72-80 detail main experiments of Figure 1 and 2. The information seems to be highly intermingled and its confusing what the approach was and what was the result. Rather than jumping between experiments it might be worthwhile to focus on Figure 1 and what it shows about *inc* expression. Figure 1 images are small and don't look very high resolution and it appears that the expression is widespread and not restricted to MB and EB as claimed by authors in the figure legend.

Figure 1 needs to be larger and if the researchers used a counter stain like nc82 that stain should also be shown as it allows neuropil detection. Nc82 stain MB and EB very clearly so the staining overlap will strengthen the authors argument.

Line 86-92: "*inc* influences sleep through adult-specific or developmental mechanisms". This is an elegant approach to address a developmental role of the gene and the schematic in Figure 2 is helpful but the rational for choose specific development stages should be expanded on as its central to the paper's conclusions. For example, why was quinic acid was not applied during individual larvae stages or two stages at a time etc. How were the time points chosen and tightly controlled between flies that could be developing at slightly different rates?

In Figure 2-5 there is extensive sleep data but only total sleep is shown. The sleep analysis involves several other features like daytime sleep, nightime sleep, sleep bout structure, activity and latency. These sleep parameters are also important and *inc* could be involved in these as well. These data should be added for all sleep experiments. This would require some data mining but no additional experiments but would be extremely useful to readers as these genes could have more specific effects on these features.

In almost all figures range of n's is provided. For example, in Figure 2 of conditional rescue N=11-86 this is quite the range so how was statistical power considered in these data sets. If there is a specific reason for why some samples were low (n-11) and so as high as 86.

Line 106-108: the authors results show that *inc* mediated circuit development is happening around third instar and this stage is critical. Why is this stage critical? What regions develop during this time and what does molecularly *inc* really do to these circuits. This needs to be discussed as it is a key conclusion of Figures 1-3.

Line 113: 283 Gal4 drivers were chosen based on what criteria? The criteria for screening are important here as there is no description of the 283 lines in methods and the data is represented in a way that not much can be concluded.

Line 124-128: Authors use *inc-Gal4*; *mb-Gal80*. MB is a structure with over 3000 neurons (KCs, ONS and DANS). The authors should make it explicit and clear as what *mb-gal80* really blocks (KCS?).

In all figures, sleep profiles should show error bars (mean and SEM).

Figure 5: Please show all sleep parameters. While total sleep data is used here to reach conclusions other features that change for manipulating different cell types in *inc* + and *inc* – background will be very interesting.

Figure 6 data is extremely interesting and really ties in the developmental role with behavior. I am surprised why they chose dfb and not EB even though *inc* expression is high in eb. Dfb is sleep-promoting but not sure how it relates to *inc* function.

The MB Gal4 drivers used in behavioral experiments are distinct from neurogenesis experiments. Reasons for doing so should be made explicit.

In all figures number of samples and statistical measures (F value, type of statistic etc.) should be stated.

Rather than using pan-MB gal4, DPM-gal4 etc. please state the name of driver in the figures and results. Several gal4 lines could have a DPM neuron and many other neurons so it won't be right to call that a DPM Gal4. This is potentially misleading. The gal4 names should be consistent and specific as they can alter the conclusions of the study.

*Reviewer #2:*

The results are very well presented and discussed. I have only 2 suggestions.

First, in the end, it seems that the authors suggest that while the *inc* mutation results in an increased number of MB neurons, these cells are hypofunctional because their axons are underdeveloped. In the limit, they may be not functional at all. If so, however, the *inc* mutation and the short sleep phenotype due to the ablation of MB – after larvae are fed with hydroxyurea – may not be very different. Is this the case? A comparison between these two short-sleeping phenotypes should be done, even just using ablation data already published.

Second, previous studies from the same authors have shown a very interesting role for *inc* in presynaptic homeostatic potentiation. I understand that this may be very difficult to study outside the neuromuscular junction, but the authors should at least try to comment on whether they think that given the postsynaptic role of *inc* in this phenomenon, the lack of this in the *inc* mutants may underlie the structural defects seen in the MB.

*Reviewer #3:*

Li et al., investigate the relationship between the *insomniac* (*inc*) gene's role in neural circuit development and sleep regulation in adult *Drosophila* using a combination of sophisticated genetics, behavioral analyses, neural circuit mapping and immunohistochemical-based measurements of neuronal structural changes. The authors find that adult sleep depends on *inc* facilitating proper wiring of the mushroom body (MB), but not other established arousal-controlling loci, during a critical window during late development that excludes adulthood. This study stands out for addressing a possibility that is often ignored in behavioral genetics (especially the field of sleep research) – namely that a gene's function may be more related to establishment of neural circuitry that controls a given behavior than acute regulation of neuronal function while the behavior is being performed. As the authors point out, their findings may also be relevant to poorly understood neurodevelopmental disorders. For example, the wiring deficiencies of *inc* mutants are consistent with *inc*'s molecular function as an adaptor of Cul3, for which human mutations have been implicated in autism.

This is a very straightforward, clear manuscript. The experiments are well-defined, rigorously performed with appropriate controls, and support the authors' conclusions quite well.

1) Lines 223-225: "Our results… suggest that many genes influence sleep through unappreciated developmental mechanisms." This statement struck me as overly bold and should probably be toned down. The findings in this study suggest that the research community should more rigorously test whether identified sleep-regulating genes function developmentally rather than assuming those functions are restricted to adult behavior. However, this study does not provide evidence that many genes have actually been misassigned adult-specific roles in regulating sleep.

2) Line line 241: "While our findings suggest that the MB is not the sole brain structure through which *inc* impacts sleep…" Presumably this statement relates to only partial suppression by *MB-Gal80* of *inc*^*1*^ rescue by *inc-Gal4/UAS-inc* Please clarify.

3) The present study cites Pfeiffenberger and Allada (Plos Genetics, 2012). However, it does not state that those authors previously showed that *inc* and *Cul3* are required during development. That omission should be corrected.

4) The authors may not want to mention this point, but I thought it was interesting that stimulation of all tested arousal circuits except MBs elicited normal (expected) changes in sleep in *inc* mutants. That suggests to me that MBs function either presynaptically or in parallel to those other sleep-regulating circuits.

---

## [Author Response]

Essential Revisions:1) The authors should briefly discuss *inc*'s previously described role in homeostatic plasticity, which may be relevant to this manuscript. Perhaps MBs require homeostatic plasticity to promote sleep but they cannot undergo it without *inc*. In that case, multiplication and rewiring of MB neurons may not be cell-autonomous direct effects of reducing *inc*'s molecular function but instead may be compensatory responses (i.e. expansion and rewiring of sleep-promoting neurons) to make up for loss of sleep that begins to manifest in larvae. Hypofunctionality of MB neurons in this context might thus be expected to reduce sleep just like loss of MB neurons following ablation with hydroxyurea. Previous studies indicate that hydroxyurea ablation of the MBs phenocopies *inc* mutants' reduction in sleep. A comparison between these two short-sleeping phenotypes should be made and can be accomplished using ablation data already published.

In the revised Discussion, we cite the role of *inc* and Cul3 in regulating synaptic homeostasis at the larval neuromuscular junction. While the simplest hypothesis supported by our current findings is that the effects that *inc* exerts on sleep through the MB reflect alterations in MB neurogenesis and anatomy, our findings do not exclude functions of *inc* at synapses that may contribute to its developmental impact on sleep. We note in the Discussion that *inc* may engage multiple substrates as a Cul3 adaptor and impact sleep by more than one mechanism.

We have added a figure comparing the short sleep phenotypes caused by *inc* mutations and MB ablation. We reproduce short sleep phenotypes elicited by MB ablation (Pitman et al., 2006; Joiner et al., 2006) and find them to be less severe than those of *inc* mutants. Because the number and anatomy of early-born MB neurons appears unaffected by the loss of *inc*, MB function in *inc* mutants is likely to be at least partially intact. Together, these data are consistent with the idea that MB function is impaired in *inc* mutants and contributes to their sleep phenotypes.

2) The authors should add at least a few lines of commentary to the *banderuola* (*bnd*) gene. Bnd controls asymmetric cell division of sensory organ neural precursors. 5' *wake* alleles are derived from *bnd*, and these alleles reduce sleep, like LOF mutants of *inc* (Mauri et al., Curr Biol 2014; Liu et al., Neuron 2014; see also Zhang et al., PLoS Genet 2015). Those findings bolster the contention in the current study that the function of many sleep-regulating genes may be developmental in origin.

We thank the reviewer for pointing out that developmental functions of the *wake/bnd* gene may underlie or contribute to its sleep phenotypes. The revised Discussion considers this relevant and interesting possibility for *wake/bnd* and other sleep mutants.

3) There were concerns related to how the data is presented, how some conclusions are made and why some critically important data is left out. Several suggesting for improvement in these regards are in the reviewers' "Recommendations for the authors." Some important points to address:a. In Figures 1-5, sleep data is only represented as a sleep profile or total sleep. Sleep is a complex behavior and when we manipulate genes several parameters are affected. Regarding sleep parameters it would be great if authors can be consistent. For example, figure 4-supplement 1 has all important sleep parameters for the Mb sleep rescue experiment but the same parameters are not shown for other experiments (e.g. quinic acid). If there is a specific reason for presenting additional parameters for one set of experiments and not the others, then it should be made explicit. It appears that authors think these parameters are important because they show them for this one set of experiments. Also, these experiments are different from the published Neuron 72:964-976. doi:10.1016/j.neuron.2011.12.003 paper showing *inc* mutants and these manipulations could affect sleep parameters differently.

We have added supplemental figures with absolute sleep parameters for animals assessed under LD cycles at constant temperature (Figures 2-4 and new Figure 6; note Figure 1 contains only immunostaining). We have also added a supplemental figure for thermogenetic activation experiments (Figure 5) containing daily sleep profiles for genotypes not included initially. For thermogenetic activation experiments, we present (1) percent change in total sleep for the entire activation day versus the baseline day and (2) daily sleep profiles, to focus on relative changes in sleep for each genotype. This enables comparison of TrpA1 activation in different neuronal populations while controlling for variation in baseline sleep across many genotypes and the differential effects of temperature alone in wild-type animals and *inc* mutants. We believe this presentation strikes a balance between plotting an overwhelming number of absolute sleep parameters for 18 genotypes over 3 days and conveying the main result: the effect on sleep elicited by neuronal activation is altered specifically for the MB in *inc* mutants.

b. There is a screen of 283 Gal4 lines in Figure 4A. The lines are listed by Bloomington id but the genotype information is absent from material and methods (copy and paste genotype information from Bloomington). We are interested in information like 49E09-Gal4 instead of BL38692 (Bloomington id) in the Materials and methods section. The text says sleep regulatory circuits or randomly selected without any additional details. How many were random, how many were sleep regulatory and where, were they clean or mixed population?

The revised Methods and Key Resources Table contain information for screened lines, fragment identity for Gal4s from the Janelia collection, additional references for Gal4 driver expression patterns, and procedures for selecting random lines. The revised manuscript contains 277 unique lines; six lines screened in duplicate were inadvertently listed as unique in the earlier revision. For the rescue screen, all but eleven lines were randomly selected.

c. Further, the figures define several transgenic lines as pan-MB gal4, DPM-gal4 etc. (Figure 5). Several gal4 lines could have a DPM neuron and many other neurons so it potentially confusing to call each a DPM Gal4. Please make it clear to the reader what Gal4 was used and where its expressed. This can be done simply in the figure legend and in a supplementary table with appropriate references to the literature so people can see for themselves where a certain line is expressed by going to the appropriate reference. This would also help people doing similar RNAi screens.

We agree that this shorthand may overstate the specificity of Gal4 lines. The revised text clarifies that expression patterns of Gal4 drivers include neurons of interest but are not limited to them. We use short labels within Figure 5A for spatial clarity; explicit identifications of Gal4s (e.g. R69F08) are present in Figure 5B, Figure 5–supplement 1, in the relevant Methods subsection, and in the Key Resources Table with relevant references.

Reviewer #1:The submitted manuscript explores an interesting area in neuroscience as to how genes control development of circuits that regulate behavior. What do these genes do in these circuits and at what stage do they become critical. Authors focus on 3 main things:1) Where *inc* is expressed and when its required for circuit development that underlies *inc* dependent sleep regulation.2) *inc* expression in MB and how that controls sleep.3) How *inc* expression regulates development of MB-KCs and assemble into functional circuits.These findings are important for the field and experiments are appropriate and figures are generally clear. My major concerns are related to how the data is presented, how some conclusions are made and why some critically important data is left out. Figure 1-5, sleep data is only represented as a sleep profile or total sleep. Sleep is a complex behavior and when we manipulate genes several parameters are affected. These sleep parameters should be shown for all genetic and other (quinic acid) manipulations.

As noted above, we have added supplemental figures with additional sleep parameters.

The screening of 283 GAL4 drivers for *inc* expression and its affect on sleep is shown but that data is barely discussed. If the intention is not to discuss the data then its positioning in main figures should be reconsidered as it is not clear what that screen is all about.

We focus on the identification of c253 and c309 as the main result of the *inc^2^* rescue screen (Figure 2A) and include screen data in the main figures to convey the scale of the screen, the distribution of screened lines, and the rank order of rescuing hits. We have refrained from discussing Gal4 drivers we did not carefully characterize or pursue; backcrossing and additional analysis of spatiotemporal expression patterns would be required to interpret possible rescue of *inc* mutants or lack thereof.

The effect of *inc* expression on KC cell number is intriguing but there is no direct evidence that reduction in the percentage of KC population affects sleep. Although, *inc* is expressed in MBs and *inc* is involved in KC neurogenesis but how do these flies with defective KCs behave. Is it the number of KCs that regulate sleep or molecular role of *inc* in MB functioning/ physiological activity?Since this paper focusses on the role of a single gene, so KCs with *inc* knockdown could be physiologically altered suggesting that while *inc* is involved in neurogenesis it is not necessarily the reason flies sleep differently. What this gene does to circuits and connectivity is a key focus of the paper but the experiments do directly address this. what is the significance of KC number to sleep and how are specific KCs related to the *inc* phenotypes? This needs to be addressed thoroughly.

The simplest model consistent with our findings is that developmental changes in the MB in *inc* mutants, including altered neurogenesis and circuit formation, impair its effects on sleep when activated in adulthood. We acknowledge that our present data do not distinguish whether changes in KC number, anatomy, physiology, or a combination of these attributes contribute to the sleep phenotypes of *inc* mutants. Resolving these mechanistic questions is non-trivial, as experimental manipulations of one process, such as MB neurogenesis, may alter the production and postmitotic development of multiple KC subtypes. We hope to pursue the underlying mechanisms, including the role of neurogenesis and MB neuron subtypes, in future studies.

Specific CommentsIntroduction: Since gene expression is the big focus of this paper the authors need to elaborate on what specific genes have been identified and how they control sleep circuits. While, the focus is *Drosophila* sleep it might be worthwhile mentioning some genes and how they shape circuits in other animals. This is an important area in sleep research that has not received much focus but the central problem and specifically role of *inc* and *Cul3* gene need to be significantly elaborated.Results section: Lines 72-80 detail main experiments of Figure 1 and 2. The information seems to be highly intermingled and its confusing what the approach was and what was the result. Rather than jumping between experiments it might be worthwhile to focus on Figure 1 and what it shows about *inc* expression. Figure 1 images are small and don't look very high resolution and it appears that the expression is widespread and not restricted to MB and EB as claimed by authors in the figure legend.Figure 1 needs to be larger and if the researchers used a counter stain like nc82 that stain should also be shown as it allows neuropil detection. Nc82 stain MB and EB very clearly so the staining overlap will strengthen the authors argument.

The revised Introduction and Discussion add context and elaborate on the roles of *inc* and *Cul3* in neuronal development. We have edited the text describing Figures 1 and 2 and hope that the changes improve clarity. We increased the size of Figure 1, allowing better resolution of staining in the EB and MB. We note that the Figure 1 legend and prior studies (Stavropoulos and Young, 2011; Pfeiffenberger and Allada, 2012) describe prominent (but not exclusive) *inc* expression in the mushroom body, ellipsoid body, fan-shaped body, and pars intercerebralis.

Line 86-92: "*inc* influences sleep through adult-specific or developmental mechanisms". This is an elegant approach to address a developmental role of the gene and the schematic in Figure 2 is helpful but the rational for choose specific development stages should be expanded on as its central to the paper's conclusions. For example, why was quinic acid was not applied during individual larvae stages or two stages at a time etc. How were the time points chosen and tightly controlled between flies that could be developing at slightly different rates?

We first validated the Q-system in constitutive rescue and then assessed developmental- and adult-specific induction regimens. We chose the wandering third instar larval stage as the first timepoint for analysis, as animals at this stage are easily selected, sexed, and transferred to different food for induction regimens. Because experiments with third instar larvae revealed that *inc* has no measurable contribution in neurons of embryos and first and second instar larvae that impacts sleep in adulthood, we did not attempt to analyze these earlier stages separately. We focused further analysis on the restricted pulse condition to map the developmental sufficiency of *inc*. While our protocol does not exclude variation in the precise developmental age of wandering third instar larvae, any such differences are accounted for in scatterplots of individual animals in Figure 2B.

In Figure 2-5 there is extensive sleep data but only total sleep is shown. The sleep analysis involves several other features like daytime sleep, nightime sleep, sleep bout structure, activity and latency. These sleep parameters are also important and *inc* could be involved in these as well. These data should be added for all sleep experiments. This would require some data mining but no additional experiments but would be extremely useful to readers as these genes could have more specific effects on these features.

We have added additional sleep parameters in supplemental figures, as described above.

In almost all figures range of n's is provided. For example, in Figure 2 of conditional rescue N=11-86 this is quite the range so how was statistical power considered in these data sets. If there is a specific reason for why some samples were low (n-11) and so as high as 86.

*inc* mutants exhibit reduced viability in many but not all genetic backgrounds in the absence of rescue (Stavropoulos and Young, 2011). The low number of animals (n=11) in the adult-specific induction regimen reflects a low number of animals obtained in this non-rescuing condition; the high number of animals (n=86) in the vehicle condition reflects the inclusion of this induction regimen in all experiments that were pooled in Figure 2. One-way ANOVA and post-hoc tests account for statistical significance between populations of different sizes.

Line 106-108: the authors results show that *inc* mediated circuit development is happening around third instar and this stage is critical. Why is this stage critical? What regions develop during this time and what does molecularly *inc* really do to these circuits. This needs to be discussed as it is a key conclusion of Figures 1-3.

As noted in the text, proliferating neuroblasts in third instar larvae and pupae give rise to adult neurons in various regions of the nervous system including the optic lobe, mushroom bodies, other brain regions, and the ventral nerve cord (White and Kankel, 1978; Truman and Bate, 1988). Prior to adulthood, these neurons elaborate projections and form synapses with their targets; many embryonic-born neurons and circuits are simultaneously remodeled and/or incorporate newly born neurons. The revised Results and Discussion note that the developmental impact of *inc* on sleep may involve processes spanning neurogenesis and postmitotic neuronal development in multiple brain regions including the mushroom body.

Line 113: 283 Gal4 drivers were chosen based on what criteria? The criteria for screening are important here as there is no description of the 283 lines in methods and the data is represented in a way that not much can be concluded.

Almost all Gal4 lines were selected randomly; eleven represent previously characterized populations whose manipulation was shown to impact sleep. The revised Methods section adds details on the identity of screened lines and their selection.

Line 124-128: Authors use inc-Gal4; mb-Gal80. MB is a structure with over 3000 neurons (KCs, ONS and DANS). The authors should make it explicit and clear as what mb-gal80 really blocks (KCS?).

We have edited the text to clarify that *MB-gal80* is expressed chiefly in intrinsic MB neurons (Kenyon cells) during development and adulthood (Krashes et al., 2007; Pauls et al., 2010).

In all figures, sleep profiles should show error bars (mean and SEM).

All sleep profiles now include shading to represent SEM.

Figure 5: Please show all sleep parameters. While total sleep data is used here to reach conclusions other features that change for manipulating different cell types in *inc* + and *inc* – background will be very interesting.

Supplemental figure panels include additional sleep parameters, as described above.

Figure 6 data is extremely interesting and really ties in the developmental role with behavior. I am surprised why they chose dfb and not EB even though *inc* expression is high in eb. Dfb is sleep-promoting but not sure how it relates to *inc* function.

*inc* is expressed in CRZ, dFB, EB, MB, and PDF neurons, among other populations (Figure 1 of this study; Stavropoulos and Young, 2011; Pfeiffenberger and Allada, 2012; Li et al., 2017); we chose CRZ, dFB, and PDF neurons for controls for anatomical analysis because their somata are easily counted and their projections are well defined. We note in the Discussion that our findings do not exclude the possibility that *inc* regulates the development of circuits outside of the MB.

The MB Gal4 drivers used in behavioral experiments are distinct from neurogenesis experiments. Reasons for doing so should be made explicit.

Split-Gal4 drivers used for anatomical analysis of specific MB neuron subtypes do not rescue *inc* sleep phenotypes (data not shown). The lack of rescue for sparsely expressed MB drivers may reflect limited strength, breadth, spatiotemporal patterns of expression, and/or a broad requirement for *inc* within the MB; we note that the anatomical defects of *inc* mutants are not limited to one subtype of MB neurons. As noted in the revised Discussion, additional manipulations will likely be required to distinguish the precise cellular populations that underlie the sleep phenotypes of *inc* mutants.

In all figures number of samples and statistical measures (F value, type of statistic etc.) should be stated.

We have added information to figure legends including F-statistics. Statistical tests described in Methods are now also included in figure legends.

Rather than using pan-MB gal4, DPM-gal4 etc. please state the name of driver in the figures and results. Several gal4 lines could have a DPM neuron and many other neurons so it won't be right to call that a DPM Gal4. This is potentially misleading. The gal4 names should be consistent and specific as they can alter the conclusions of the study.

We have added identifications for drivers as noted above.

Reviewer #2:The results are very well presented and discussed. I have only 2 suggestions.First, in the end, it seems that the authors suggest that while the *inc* mutation results in an increased number of MB neurons, these cells are hypofunctional because their axons are underdeveloped. In the limit, they may be not functional at all. If so, however, the *inc* mutation and the short sleep phenotype due to the ablation of MB – after larvae are fed with hydroxyurea – may not be very different. Is this the case? A comparison between these two short-sleeping phenotypes should be done, even just using ablation data already published.

We thank the reviewer for these comments and helpful suggestions. The revised manuscript includes comparative analysis of the sleep phenotypes caused by *inc* mutations and MB ablation.

Second, previous studies from the same authors have shown a very interesting role for *inc* in presynaptic homeostatic potentiation. I understand that this may be very difficult to study outside the neuromuscular junction, but the authors should at least try to comment on whether they think that given the postsynaptic role of *inc* in this phenomenon, the lack of this in the *inc* mutants may underlie the structural defects seen in the MB.

The revised discussion cites the role of *inc* and Cul3 in synaptic homeostasis at the larval neuromuscular junction and notes that developmental functions of *inc* at synapses may contribute to *inc* sleep phenotypes. Although we are unaware of instances in which defects in synaptic homeostasis elicit compensatory alterations in neurogenesis or neuronal projections in a cell autonomous or non-cell autonomous manner, we cannot at present exclude this possibility for *inc*. As noted in the revised Discussion, the identification of Inc substrates that impact sleep may provide insight into the downstream cellular and molecular pathways.

Reviewer #3:Li et al., investigate the relationship between the *insomniac* (*inc*) gene's role in neural circuit development and sleep regulation in adult *Drosophila* using a combination of sophisticated genetics, behavioral analyses, neural circuit mapping and immunohistochemical-based measurements of neuronal structural changes. The authors find that adult sleep depends on *inc* facilitating proper wiring of the mushroom body (MB), but not other established arousal-controlling loci, during a critical window during late development that excludes adulthood. This study stands out for addressing a possibility that is often ignored in behavioral genetics (especially the field of sleep research) – namely that a gene's function may be more related to establishment of neural circuitry that controls a given behavior than acute regulation of neuronal function while the behavior is being performed. As the authors point out, their findings may also be relevant to poorly understood neurodevelopmental disorders. For example, the wiring deficiencies of inc mutants are consistent with *inc*'s molecular function as an adaptor of Cul3, for which human mutations have been implicated in autism.This is a very straightforward, clear manuscript. The experiments are well-defined, rigorously performed with appropriate controls, and support the authors' conclusions quite well.1) Lines 223-225: "Our results… suggest that many genes influence sleep through unappreciated developmental mechanisms." This statement struck me as overly bold and should probably be toned down. The findings in this study suggest that the research community should more rigorously test whether identified sleep-regulating genes function developmentally rather than assuming those functions are restricted to adult behavior. However, this study does not provide evidence that many genes have actually been misassigned adult-specific roles in regulating sleep.

This point is well taken and we have adjusted the language of this section.

2) Line line 241: "While our findings suggest that the MB is not the sole brain structure through which *inc* impacts sleep…" Presumably this statement relates to only partial suppression by *MB-Gal80* of *inc*^*1*^ rescue by *inc-Gal4/UAS-inc*. Please clarify.

This is correct. The ability of *MB-gal80* to partially suppress the rescue conferred by *inc-Gal4* suggests that the mushroom body is a critical region mediating the effects of *inc* on sleep, among other neuronal populations. We have edited the text to clarify this point.

3) The present study cites Pfeiffenberger and Allada (Plos Genetics, 2012). However, it does not state that those authors previously showed that *inc* and *Cul3* are required during development. That omission should be corrected.

The Introduction references earlier publications (Pfeiffenberger and Allada, 2012 and Li and Stavropoulos, 2016) relevant to the temporal requirements of *inc* with regard to sleep regulation. Pfeiffenberger and Allada (2012) reported developmental manipulations of *inc* using the Geneswitch system, but we were unable to reproduce their findings (Li and Stavropoulos, 2016). In particular, we found that developmental induction of Geneswitch in neurons causes lethality and strong nonspecific changes in sleep in the absence of effector transgenes or with *UAS-GFP* (Li and Stavropoulos, 2016). We believe these experimental artifacts preclude use of the Geneswitch system for developmental manipulations of sleep, and thus we regard the temporal requirements of *inc* as unresolved prior to the current study. In contrast, Q-system transgenes and their induction with quinic acid have no measurable effect on sleep in the absence of effector transgenes in wild-type animals or *inc* mutants, as shown in the present study and as reported previously (Li and Stavropoulos, 2016).

4) The authors may not want to mention this point, but I thought it was interesting that stimulation of all tested arousal circuits except MBs elicited normal (expected) changes in sleep in *inc* mutants. That suggests to me that MBs function either presynaptically or in parallel to those other sleep-regulating circuits.

Recent connectome data for the mushroom body provide information on its direct and indirect synaptic connectivity (Li et al., 2020, *eLife*; Schlegel et al., 2021, *eLife*), including to fan-shaped body neurons in the central complex. MB output neurons (MBONs) provide direct and indirect presynaptic input to many neurons of the fan-shaped body (Li et al., 2020, *eLife*). While no direct inputs from fan-shaped body neurons to the MB are apparent from connectome data, some fan-shaped body neurons provide direct input to dopaminergic neurons (DANs), which can modulate synapses between MB neurons and MBONs. Thus, FB neurons may indirectly modulate MB output. The functional relevance of these synaptic connections in the context of sleep remains unknown and awaits combinatorial manipulations of circuit elements; we note that residual MB function in *inc* mutants might not fully block potential presynaptic input from neuronal populations in our analysis.